# Identifying in vivo genetic dependencies of melanocyte and melanoma development

Sarah Perlee[1,2], Yilun Ma[1,3,4], Miranda V Hunter[1], Jacob B Swanson[5,6], Nelly M Cruz[1], Zhitao Ming[7], Julia Xia[1], Timothee Lionnet[5,6,8], Maura McGrail[7], Richard M White[1,9]*

[1]Department of Cancer Biology and Genetics, Memorial Sloan Kettering Cancer Center, New York, United States; [2]Gerstner Sloan Kettering Graduate School of Biomedical Sciences, Memorial Sloan Kettering Cancer Center, New York, United States; [3]Weill Cornell/Rockefeller/Sloan Kettering Tri-Institutional MD-PhD Program, Memorial Sloan Kettering Cancer Center, New York, United States; [4]Cell and Developmental Biology Program, Weill Cornell Graduate School of Medical Sciences, New York, United States; [5]Institute for Systems Genetics, NYU Grossman School of Medicine, New York, United States; [6]Department of Cell Biology, NYU Grossman School of Medicine, New York, United States; [7]Department of Genetics, Development and Cell Biology, Iowa State University, Ames, United States; [8]Department of Biomedical Engineering, NYU Tandon School of Engineering, Brooklyn, United States; [9]Nuffield Department of Medicine, Ludwig Institute for Cancer Research, University of Oxford, Oxford, United Kingdom

*For correspondence:
richard.white@ludwig.ox.ac.uk

## eLife Assessment

This **important** manuscript introduces a genetic tool utilizing mutant mitfa-Cas9 expressing zebrafish to knockout genes to analyze melanocyte function in development and tumorigenesis. The data are **convincing** and the authors cover potential caveats from their model that might impact its utility for future work. This work significantly adds to the existing approaches in the field, as the mitfa:Cas9 strategy taken here provides a roadmap for generating similar platforms for using other tissue-specific regulators and Cas proteins in the future.

**Abstract** The advent of large-scale sequencing in both development and disease has identified large numbers of candidate genes that may be linked to important phenotypes. We have developed a rapid, scalable system for assessing the role of candidate genes using zebrafish. We generated transgenic zebrafish in which Cas9 was knocked in to the endogenous *mitfa* locus, a master transcription factor of the melanocyte lineage. The main advantage of this system compared to existing techniques is maintenance of endogenous regulatory elements. We used this system to identify both cell-autonomous and non-cell-autonomous regulators of normal melanocyte development. We then applied this to the melanoma setting to demonstrate that loss of genes required for melanocyte survival can paradoxically promote more aggressive phenotypes, highlighting that in vitro screens can mask in vivo phenotypes. Our genetic approach offers a versatile tool for exploring developmental processes and disease mechanisms that can readily be applied to other cell lineages.

## Introduction

Cell-type-specific knockout of genes is crucial for unraveling the intricate molecular mechanisms underlying cell function, development, and transformation. Efficient methods are critical as sequencing projects such as All of Us, the UK Biobank, and the Cancer DepMap continue to identify hundreds of new candidate genes on a regular basis, many of which have yet to be characterized in vivo (*Mayer and Huser, 2023*; *Sudlow et al., 2015*; *Tsherniak et al., 2017*). Zebrafish (*Danio rerio*) have emerged as a powerful genetic model organism for studying these processes. The large number of animals that can be easily generated makes it advantageous (compared to other systems like mice) for querying large numbers of genetic variants that may have links to human diseases. However, most studies in this system use germline knockout, which makes it difficult to determine whether the identified gene has a specific function in a particular lineage, or whether the phenotype reflects a more general developmental defect. Furthermore, many genes that might have lineage-specific functions cannot be studied using germline knockout, since embryonic lethality often accompanies global knockout of essential genes (*Amsterdam et al., 2004*). To overcome these limitations, several groups have explored tissue-specific and conditional knockout strategies in zebrafish (*Yin et al., 2015*; *Ablain et al., 2015*; *Zhou et al., 2018*; *Ni et al., 2012*; *Grajevskaja et al., 2018*; *Burg et al., 2018*; *Kalvaitytė and Balciunas, 2022*; *Singh Angom et al., 2023*; *Shin et al., 2023*). For example, the MAZERATI system uses tissue-specific promoter transgenes to drive Cas9 expression, which are introduced via plasmid injection into one-cell stage embryos (*Ablain et al., 2015*; *Ablain et al., 2021*). This system has been shown to be versatile and powerful. One potential limitation of promoter fragment transgenes is that these fragments could lack certain regulatory regions. For this reason, it is common in systems such as mice to knock in genetic cassettes (i.e. Cre recombinase) into endogenous loci.

In this study, we wished to develop such an endogenous system to understand in vivo genetic dependencies, using melanocytes and melanoma as an exemplar. Skin color pigmentation is among the most heterogeneous of genetic systems, with a large number of loci linked to melanocyte development. Mutations in pleiotropic genes such as *sox10, ednrb, jam3b, tuba8l3, meox1,* and *kit* yield clear defects in melanocyte/melanophore development, but these mutants also have defects in other tissues (*Dutton et al., 2001*; *Parichy et al., 2000*; *Eom et al., 2021*; *Parichy and Turner, 2003*; *Nguyen et al., 2014*; *Irion et al., 2016*; *Parichy et al., 1999*). Many of these genes are also expressed in melanoma, but their specific in vivo function in tumorigenesis remains poorly understood. By integrating the Cas9 nuclease into the endogenous melanocyte-specific *mitfa* locus, we achieved specific gene disruption across the melanocyte developmental lineage. We used our *mitfa*$^{Cas9}$ knock-in animals to inactivate the pigmentation gene *albino* and the embryonic essential genes *sox10, tuba1a,* and *ptena/b* in melanocytes without additional developmental or off-target phenotypes. We also demonstrated that this system can be used to induce melanomas in wild-type (WT) fish by inactivating the tumor suppressors *tp53* and *ptena/b* within melanocytes that also express the melanoma oncogene BRAF$^{V600E}$. Inactivating the neural crest transcription factor *sox10* in these tumors significantly decreased melanoma initiation but resulted in rare invasive tumors that highly expressed the *sox9* transcription factor, highlighting that unexpected in vivo phenotypes can emerge that aren't predicted from in vitro studies (*Capparelli et al., 2022*; *Wouters et al., 2020*). Our system can thus be used to advance our understanding of melanocyte biology and the genetic underpinnings of phenotypic switching in melanoma, an approach that can readily be used for other cell types.

## Results

### Targeted integration of Cas9 to the *mitfa* locus

To selectively disrupt genes in a melanocyte lineage-specific manner while retaining natural regulatory elements, we established a transgenic zebrafish line harboring Cas9 at the endogenous *mitfa* locus. *Mitfa* expression is detectable by 18 hr post-fertilization (hpf) and is highly expressed in pigment cells, including melanocytes, xanthophores, and their progenitors. To achieve knock-in, we utilized the GeneWeld method, which enables efficient integration of DNA cassettes using homology-mediated end joining (*Welker et al., 2021*; *Wierson et al., 2020*). The knock-in cassette, containing Cas9 and BFP driven by the eye-specific γ-crystallin promoter, was targeted to exon 2 of the *mitfa* gene, which leads to disruption of the endogenous *mitfa* gene (*Figure 1A*). We strategically selected this genomic site for several reasons. First, this is the sole exon shared among the three *mitfa* protein-coding

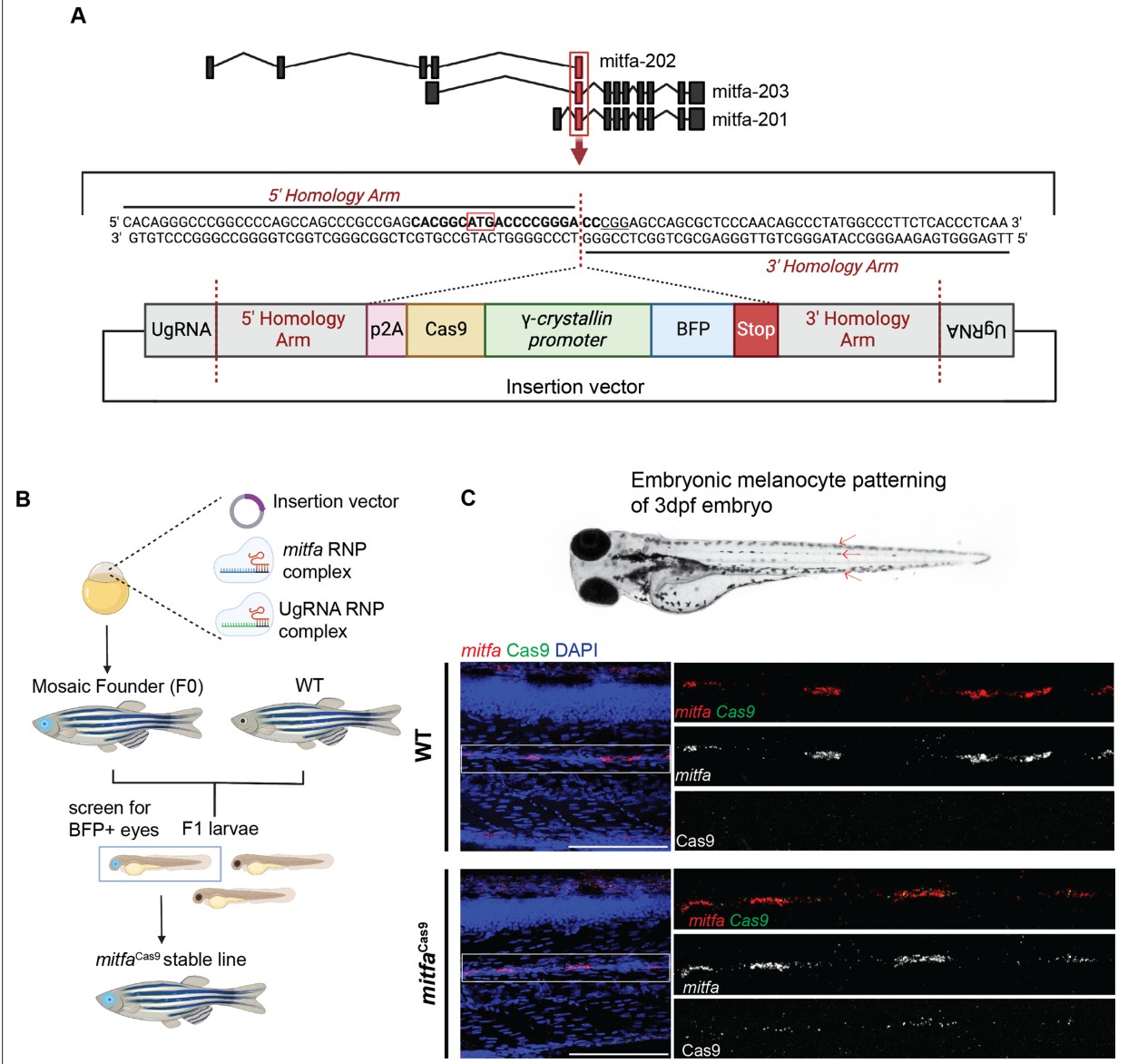

**Figure 1.** Targeted integration of Cas9 to the *mitfa* locus with GeneWeld knock-in system. (**A**) Schematic depicting integration site and GeneWeld knock-in cassette. 5' and 3' homology arms are designed to target exon 2 of the zebrafish *mitfa* gene. The knock-in cassette includes p2A followed by Cas9 and BFP driven by the eye-specific promoter γ-crystallin. Created with BioRender.com. (**B**) Pipeline to produce F1 *mitfa*Cas9 zebrafish. The GeneWeld knock-in vector and gRNAs targeting the *mitfa* genomic insertion site or specific sites on the knock-in vector (UgRNA) are co-injected into one-cell-stage wild-type (WT) zebrafish embryos. Embryos are screened for BFP+ eyes marking mosaic integration. These mosaic founder fish are then raised to adulthood and crossed with WT fish. The resulting embryos are screened for BFP+ eyes and sequenced to confirm precise integration. Created with BioRender.com. (**C**) Fluorescence in situ hybridization (FISH) chain reaction on 3 days post-fertilization (dpf) WT and *mitfa*Cas9 embryos treated with 1-phenyl 2-thiourea (PTU). Arrows on the whole embryo image (WT 3 dpf embryo not treated with PTU) indicate the embryonic melanocyte stripe regions where *mitfa* and Cas9 expression is expected. The presence of *mitfa* and Cas9 RNA was assessed by confocal microscopy at 40x magnification. Maximum intensity projections are shown. n=17 WT embryos and n=16 *mitfa*Cas9 embryos were screened. Scale bars, 100 µm.

The online version of this article includes the following figure supplement(s) for figure 1:

**Figure supplement 1.** Validation of Cas9 knock-in.

transcripts. Furthermore, targeting Cas9 to exon 2 offers the flexibility of replicating this knock-in strategy in *casper* zebrafish, a transparent fish widely used for melanoma studies that harbor a premature stop codon in *mitfa* exon 3 (**White et al., 2008**).

The knock-in vector, along with Cas9 protein and guide RNAs targeting either the *mitfa* integration site or the sequences flanking the 5' and 3' homology arms, was injected into one-cell stage WT

Tropical 5D (T5D) zebrafish embryos (**Figure 1B**; **Balik-Meisner et al., 2018**). An average of 19% of injected embryos had detectable ocular BFP expression at 3 days post-fertilization (dpf), indicating successful integration (**Figure 1—figure supplement 1A**). Embryos were raised to adulthood, then crossed with WT T5D fish. The resulting F1 embryos were screened for BFP+ eyes, and tail clippings were sequenced to confirm precise integration of the knock-in cassette at the *mitfa* locus. We sequenced four F1 fish and detected on-target integration at the *mitfa* locus in all four fish (**Figure 1—figure supplement 1B**). In zebrafish, one copy of *mitfa* is sufficient for the normal development of neural crest-derived melanocytes, but loss of both copies of *mitfa* results in loss of the melanocyte lineage (**White et al., 2008**; **Lister et al., 1999**). To determine the effect of a single copy of *mitfa*[Cas9] on melanocytes, *mitfa*[Cas9] fish were in-crossed, and the resulting siblings were imaged at various time points (**Figure 1—figure supplement 1C**). WT and *mitfa*[Cas9] fish exhibited indistinguishable melanocyte patterning during both embryonic and adult stages. However, as expected, fish carrying two copies of Cas9 (*mitfa*[Cas9/Cas9]) resembled the *nacre* mutant (*mitfa-/-*), characterized by complete loss of melanocytes, indicating the endogenous *mitfa* gene is disrupted on knock-in alleles (**Lister et al., 1999**). We used zebrafish harboring one knock-in allele (*mitfa*[Cas9]) for all subsequent experiments to allow us to test gene function in melanocytes.

To determine whether Cas9 expression was restricted to melanocytes, we performed fluorescence in situ hybridization (FISH) on 3 dpf WT and *mitfa*[Cas9] embryos to assess *mitfa* and Cas9 mRNA expression (**Figure 1C**, **Figure 1—figure supplement 1D**). As expected, *mitfa* was detected within the embryonic melanocyte stripe regions in both WT and *mitfa*[Cas9] fish. In *mitfa*[Cas9] embryos, Cas9 expression was observed in an average of 86% of these *mitfa*+ cells (**Figure 1—figure supplement 1D**). These results provide functional validation for on-target integration and melanocyte lineage-restricted expression of Cas9.

## Melanocyte-specific loss of the human skin color-associated gene *slc45a2* results in robust loss of pigmentation

To assess the effectiveness of our stable *mitfa*[Cas9] line, we first targeted the *albino* gene (*slc45a2*). Large-scale GWASs have repeatedly identified SNPs in this gene as tightly linked to human skin color variation, and germline disruption leads to a complete loss of pigmentation in melanocytes (**Fernandez et al., 2008**; **Branicki et al., 2008**; **Streisinger et al., 1986**). This distinct phenotype enables the visual observation of Cas9 activity in zebrafish injected with *albino* gRNA. To test our system, we designed plasmids, denoted MG-gRNA, containing a zU6:gRNA cassette followed by *mitfa*:GFP to enable visualization of cells expressing the *albino* gRNA. MG-*albino* and MG-NT were injected into one-cell-stage embryos from crosses between *mitfa*[Cas9] and WT fish, which express Cas9 in all melanocytes but exhibit mosaic gRNA expression (**Figure 2A**).

Embryos were screened for BFP+ eyes to identify those harboring *mitfa*[Cas9]. Upon adulthood, 92% of F0 MG-*albino* fish had mosaic loss of pigmentation within the melanocyte stripe regions (**Figure 2B and C**). Unpigmented regions maintained GFP expression (**Figure 2B**), indicating that the breaks in melanocyte stripes arise from loss of pigmentation in melanocytes rather than loss of the melanocyte lineage (**Figure 2B**). As expected, control MG-NT clutch mates displayed normal pigmentation, and GFP+ melanocytes were indistinguishable from GFP- regions on the same fish (**Figure 2B and C**).

We next aimed to determine the efficiency of *mitfa*[Cas9] in the F1 generation, in which every melanocyte stably expresses the *albino* gRNA and *mitfa*[Cas9]. To generate F1 lines, we outcrossed the F0 MG-gRNA fish to WT and sorted embryos for BFP+ eyes and GFP+ melanocytes. F1 MG-*albino* fish exhibited a near-complete loss of pigmentation, demonstrating that our *mitfa*[Cas9] system is highly efficient in F1 fish (**Figure 2D and E**). To study the effect of *albino* inactivation across various stages of development, F1 MG-gRNA fish were once again outcrossed to WT, generating F2 MG-gRNA fish expressing one copy of *mitfa*[Cas9]. Imaging revealed a measurable pigmentation loss in F2 MG-*albino* melanocytes as early as 3 dpf (**Figure 2F and G**). F2 MG-*albino* fish maintained loss of pigmentation throughout development, indicating the sustained activity of *mitfa*[Cas9] (**Figure 2—figure supplement 1A–D**).

To confirm on-target cutting of the *albino* gene in the melanocyte lineage (and not other tissues), the skin from F1 MG-*albino* fish was dissected and GFP+ and GFP- cells were isolated using FACS (**Figure 2H**). GFP+ cells encompass *mitfa*-expressing melanocytes, xanthophores, and their progenitors, while GFP- cells represent other non-melanocytic, non-*mitfa*-expressing skin cells, such as

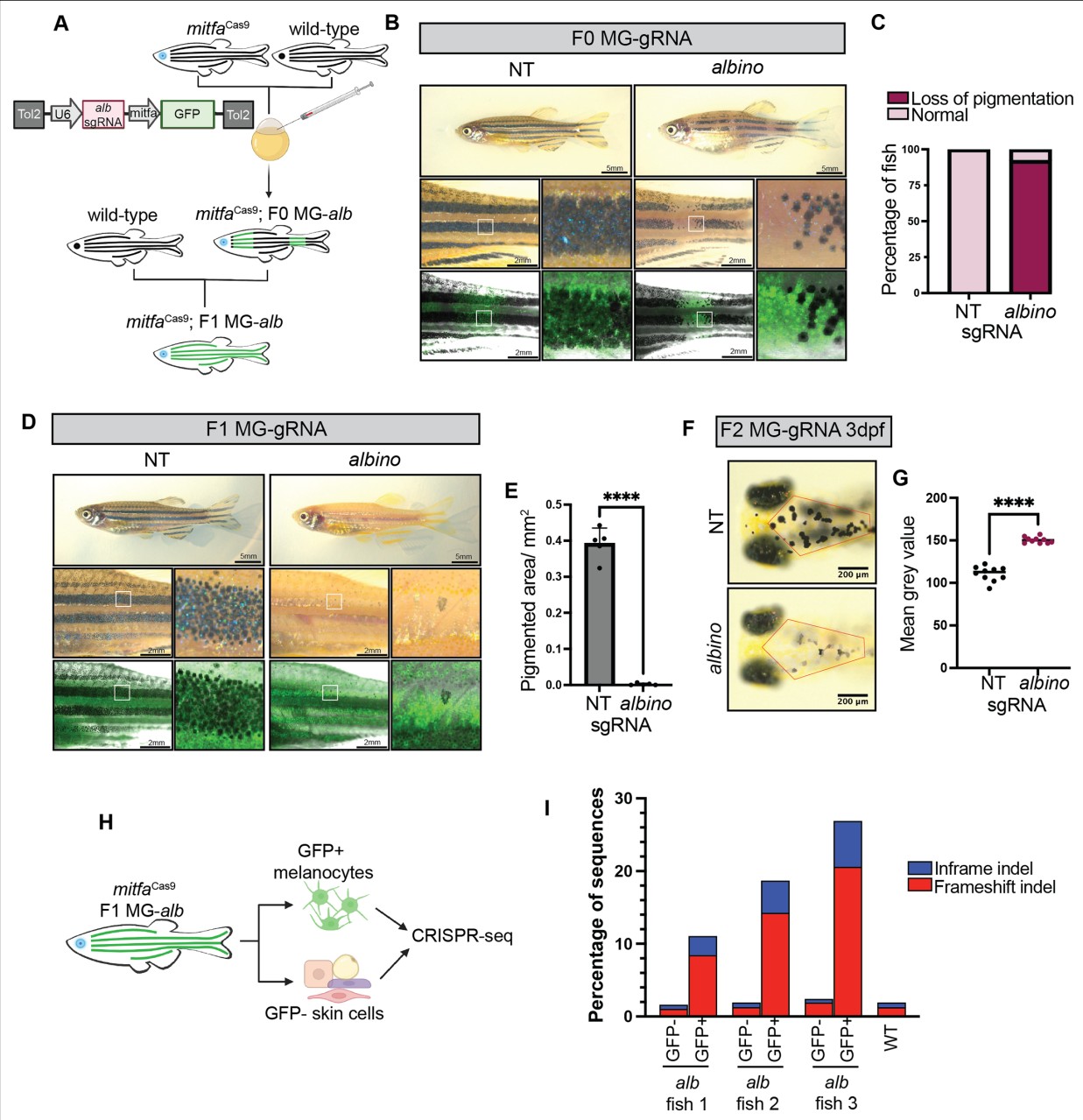

**Figure 2.** Melanocyte lineage-specific knockout of *albino* using *mitfa*^Cas9 fish. (**A**) Pipeline to generate F0 and F1 U6:gRNA; *mitfa*:GFP (MG-gRNA) zebrafish. Created with BioRender.com. (**B**) Adult *mitfa*^Cas9 MG-NT and MG-*albino* F0 fish. (**C**) Proportion of MG-NT (n=19) and MG-*albino* (n=25) F0 fish with loss of pigmentation phenotype. (**D**) Adult *mitfa*^Cas9 MG-NT and MG-*albino* F1 fish. (**E**) Pigmented area/mm² calculated for n=5 fish/genotype within a defined rectangular region of interest (ROI) encompassing the top and middle melanocyte stripes. Two-sided Student's t-test was used to assess statistical significance, ****p<0.0001; error bars, SD. (**F**) 3 days post-fertilization (dpf) *mitfa*^Cas9 MG-NT and MG-*albino* F2 fish. Representative embryos are shown for each genotype. (**G**) Mean gray value of head melanocytes calculated for n=10 embryos/genotype within a defined hexagonal ROI indicated as a red outline in **F**. Two-sided Student's t-test was used to assess statistical significance, ****p<0.0001. (**H**) Schematic for CRISPR sequencing protocol. Created with BioRender.com. (**I**) CRISPR-seq results are shown for WT and n=3 independent F1 MG-*albino* fish. Only GFP-negative cells were isolated from WT fish. Results are shown as a fraction of sequences with indels calculated using CRISPResso.

The online version of this article includes the following figure supplement(s) for figure 2:

**Figure supplement 1.** *albino* knockout fish.

keratinocytes, fibroblasts, and immune cells. We performed CRISPR-seq on DNA from both populations to directly measure Cas9 efficiency and cell-type specificity, quantifying the proportion of indels at the *albino* locus. Although we have previously shown that FACS melanocytic cells are possible, it is important to note that we often observe high levels of contaminating keratinocytes in FACS-sorted melanocyte populations, which may lead to an underestimation of the true allelic frequency in our CRISPR-seq (*Weiss et al., 2022a*). Despite this, robust inactivation of the *albino* gene was observed in GFP+ (*mitfa*-expressing) cells, confirming somatic inactivation (*Figure 2I*). The majority of indels were frameshift-inducing mutations. A likely PCR or sequencing artifact resulting in a 'T' insertion was excluded from the frameshift indel percentage as it was observed in all conditions, including WT fish with no pigmentation phenotype (*Figure 2—figure supplement 1E*). These results confirm that our *mitfa*$^{Cas9}$ system allows us to inactivate genes within the melanocytic lineage in vivo in a cell-type-specific manner.

## The neural crest-related gene *sox10* has specific function in adult melanocyte patterning and regeneration

Having confirmed that our *mitfa*$^{Cas9}$ system allows us to manipulate melanocyte gene expression, we next wanted to target embryonic essential genes. Detecting 'negative' gene knockout phenotypes such as cell death can be challenging in vivo due to the selective advantage that WT cells have over mutants.

To assess the effectiveness of our *mitfa*$^{Cas9}$ model in detecting negative phenotypes, we utilized it to target *sox10*, a transcription factor vital to the neural crest lineage and indispensable for overall survival of both zebrafish and mice (*Dutton et al., 2001*; *Hou et al., 2006*). Consequently, a global knockout approach is not feasible for studying the function of *sox10* in adult melanocytes in vivo. Germline knockout of *sox10* in zebrafish leads to severe defects in peripheral nerve development and the complete absence of melanocytes, and the animals die around day 14 (*Dutton et al., 2001*; *Kelsh et al., 1996*; *Carney et al., 2006*). Due to the complete absence of melanocytes and their precursor cells, melanoblasts, in *sox10*$^{-/-}$ fish, it has not been possible to study the specific role of *sox10* on these more differentiated cell types in vivo. One of the key unanswered questions is how *sox10* balances its role in maintaining melanoblast stemness with promoting terminal differentiation, particularly in the context of adult melanocyte regeneration.

We leveraged our *mitfa*$^{Cas9}$ system to specifically inactivate *sox10* in the melanocyte lineage. *mitfa*$^{Cas9}$ fish injected with an MG-*sox10* plasmid exhibited noticeable gaps in their melanocyte stripes (*Figure 3A and B*). Germline deletion of a *sox10* enhancer region similarly affects adult stripe patterning in zebrafish (*Cunningham et al., 2021*). In contrast to the *albino* F0 fish, the gaps in the stripes of the *sox10* F0 fish were not occupied by unpigmented GFP+ melanocytes, indicating a potential defect in the survival or differentiation of melanocytes upon loss of *sox10*. This phenotype became even more pronounced in the stable F1 lines, where several gaps in the melanocyte stripes are visible (*Figure 3C*). Zebrafish have four dark stripes occupied by melanocytes, which alternate with light-colored interstripes occupied by yellow-pigmented xanthophores (*Owen et al., 2020*). Typically, melanocyte stripe 1D (dorsal) is wider than the interstripe X0 directly below it, as observed in MG-NT F1 fish (*Figure 3C and D*). However, the opposite is true for MG-*sox10* F1 fish, where we observed an apparent widening of the interstripe region and narrowing of the melanocyte stripes (*Figure 3C and D*). Melanocytes were counted in fish treated with epinephrine to aggregate melanosomes, revealing a significant reduction in the number of melanocytes in the F1 *sox10* KO fish compared to F1 NT fish (*Figure 3E*).

Melanocytes continually regenerate throughout life, typically from a progenitor population of melanoblasts or melanocyte stem cells. We next assessed whether *sox10* was required for this. We treated adult fish with neocuproine, a copper chelator previously shown to kill mature, pigmented melanocytes (*O'Reilly-Pol and Johnson, 2008*). We then compared regeneration from these progenitors in MG-*sox10* and MG-NT F1 fish by counting the percentage of melanocytes that emerged after 7, 15, and 70 days (*Figure 3F*). At days 7 and 15, when the melanocytes were still in the process of regenerating from melanoblasts, there were no significant differences between the two groups. However, by the time regeneration was completed, we found a significant disparity between the regenerative potential of MG-NT and MG-*sox10* fish. At 70 days post-neocuproine treatment, MG-NT fish had regenerated an average of 82% of their melanocytes, whereas only 53% of melanocytes

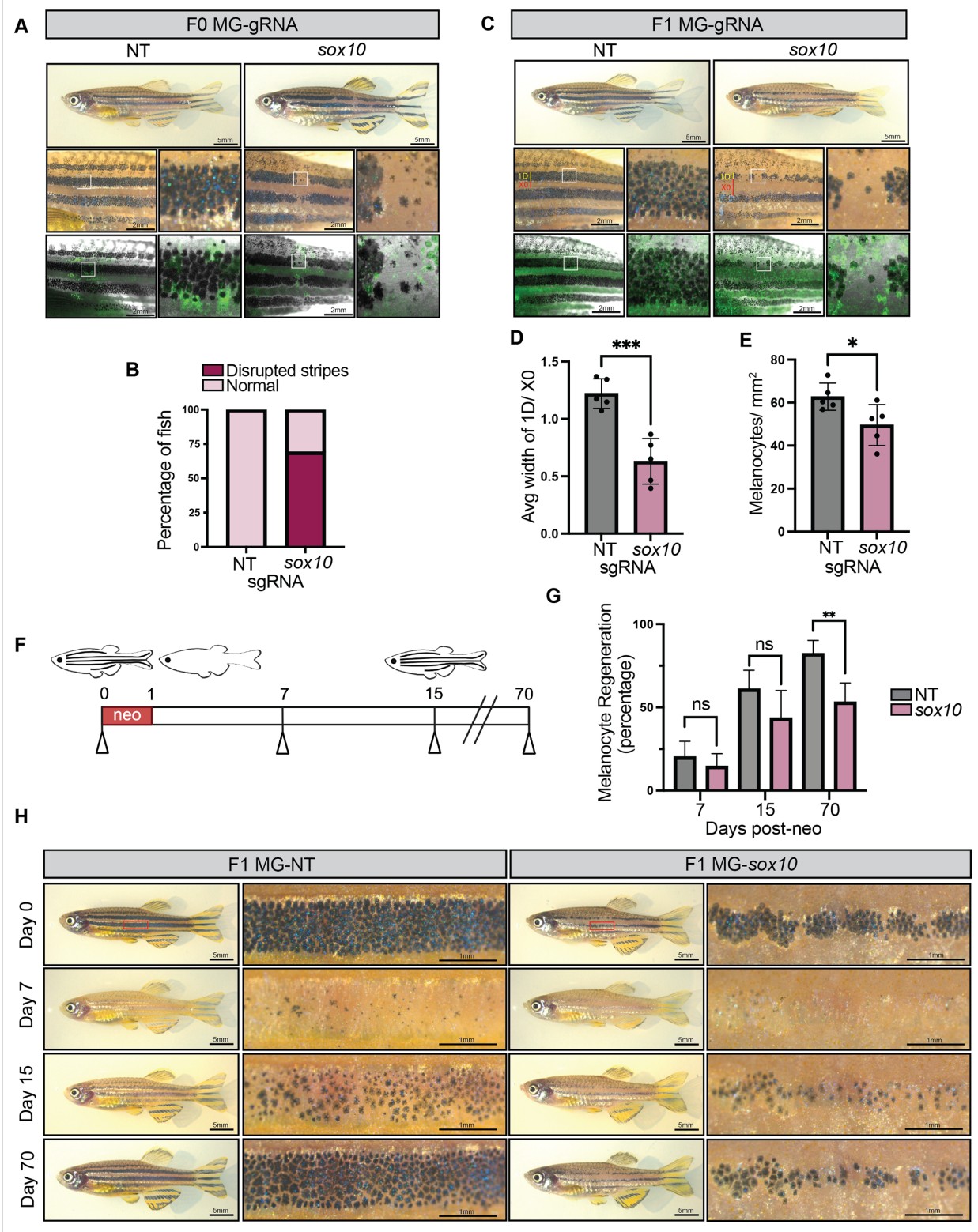

**Figure 3.** Melanocyte lineage-specific knockout of *sox10* using *mitfa*^Cas9 fish. (**A**) Adult *mitfa*^Cas9 MG-NT and MG-*sox10* F0 fish. (**B**) Proportion of MG-NT (n=19) and MG-*sox10* (n=16) F0 fish with disrupted stripes phenotype. (**C**) Adult *mitfa*^Cas9 MG-NT and MG-*sox10* F1 fish. (**D**) The average width of melanocyte stripe 1D and xanthophore interstripe X0 was calculated for each fish by averaging 5 stripes/interstripe width measurements. N=5 fish per genotype. Two-sided Student's t-test was used to assess statistical significance, ***p<0.001; error bars, SD. (**E**) F1 MG-NT (n=5) and MG-*sox10* (n=5) adult fish were treated with epinephrine, and melanocytes were counted within a defined rectangular region of interest 3.46 mm × 2.54 mm

*Figure 3 continued on next page*

Figure 3 continued

encompassing the top and middle melanocyte stripes. Two-sided Student's t-test was used to assess statistical significance, *p<0.05; error bars, SD. (**F**) Schematic of neocuproine (neo) experimental setup. Adult *mitfa*^Cas9 MG-NT (n=5) and MG-*sox10* (n=5) F1 fish were treated with neocuproine for 24 hr to ablate melanocytes, then imaged at days 7, 15, and 70 to measure regeneration of melanocytes compared to day 0. Created with BioRender.com. (**G**) Quantification of melanocyte regeneration. Fish were treated with epinephrine prior to imaging to enable counting of melanocytes. Two-sided Student's t-test was used to assess statistical significance, **p<0.01; error bars, SD. (**H**) Representative images are shown for MG-NT and MG-*sox10* fish pre- and post-neocuproine treatment.

had regenerated in MG-*sox10* fish (*Figure 3G and H*). These findings demonstrate that *sox10* is required for adult melanocyte regeneration, highlighting its requirement within melanoblast populations outside of neural crest specification.

## Non-autonomous functions of *tuba1a/tuba1c* on melanocytes

One of the significant challenges of conventional global knockout methods is the inability to differentiate between cell-autonomous and non-autonomous phenotypes. This distinction is particularly crucial in melanocyte studies due to their direct and indirect interactions with diverse cell types. A notable example of genes with pleiotropic effects is the tubulin gene family, whose α/β tubulin heterodimers form microtubules essential for various cellular processes such as cell division, motility, and intracellular transport across various cell types (*Cushion et al., 2023*). Microtubules are also critical for the transport of melanosomes in melanocytic cells (*Rogers et al., 1997*).

Mutation of tubulin genes such as *tuba8l3a* in zebrafish leads to highly pleiotropic effects, including defects in melanocyte patterning, CNS abnormalities, and altered craniofacial morphology (*Larson et al., 2010*). We chose to focus on the closely related α tubulin gene *tuba1a*, which is highly expressed in many cell types of the developing zebrafish, including melanocytes, but whose specific function in these cells remains unexplored (*Farnsworth et al., 2020*). Non-cell-type-specific knockout of *tuba1a* with the Alt-R CRISPR Cas9 system results in extensive embryo abnormalities, including a curved tail phenotype, pericardial edema, and melanocytes with more dispersed pigmentation (*Figure 4A and B*, *Figure 4—figure supplement 1A and B*). We found that compared to 71% survival of NT Alt-R embryos, only 10% of embryos injected with *tuba1a* Alt-R survive to 14 dpf (*Figure 4C*). During validation of *tuba1a* Alt-R, we discovered that this guide also targets *tuba1c*, a gene with 99.78% homology to *tuba1a* in zebrafish (*Figure 4—figure supplement 1B*). Despite robust cutting, neither *tuba1a*- nor *tuba1c*-specific guides alone were sufficient to recreate the dispersed melanocyte phenotype (*Figure 4—figure supplement 1C and D*). However, we were able to recapitulate our findings using a second guide RNA that simultaneously targeted both *tuba1a* and *tuba1c*, suggesting possible redundancy of these genes, with compensation masking the phenotype in single-gene knockouts (*Figure 4—figure supplement 1E and F*).

Similar to *sox10*, the embryonic lethality of global *tuba1a/c* knockout prevents the study of knockout phenotypes in adult fish. To overcome this limitation, we implemented our *mitfa*^Cas9 system to specifically knock out *tuba1a/c* in melanocytes (*Figure 4—figure supplement 1G and H*). CRISPR sequencing of GFP+ cells isolated from *mitfaCas9* MG-*tuba1a/c* fish confirmed successful knockout of both *tuba1a* and *tuba1c*, with knockout efficiencies comparable to those observed in MG-albino fish (*Figure 4—figure supplement 1I*). Unlike global knockout of *tuba1a/c*, melanocyte-specific loss of *tuba1a/c* resulted in viable adult fish with no obvious abnormalities (*Figure 4—figure supplement 1G and H*). Surprisingly, we did not observe the dispersed melanocyte phenotype in any *mitfa*^Cas9 MG-*tuba1a/c* embryos (*Figure 4D and E*). Based on this observation, we hypothesized that *tuba1a/c* may be functioning in a cell non-autonomous manner on melanocytes.

The dispersed pigmentation observed in *tuba1a/c* Alt-R embryos resembles that seen in blind zebrafish, which fail to contract their melanosomes in response to light (*Neuhauss et al., 1999*). This light-mediated camouflage involves retinal ganglion cells signaling hypothalamic neurons to release melanin-concentrating hormone (MCH), which then binds to MCH receptors on melanocytes, triggering melanosome aggregation via microtubules (*Neuhauss et al., 1999*). The normal melanosome dispersion observed in *mitfa*^Cas9 MG-*tuba1a/c* fish suggests that rather than a cell-intrinsic defect in melanosome aggregation, *tuba1a/c* may indirectly influence melanocytes through potential defects in retinal and/or central nervous system cells crucial for light-mediated camouflage. Interestingly, *TUBA1A* mutation in humans can result in both ophthalmologic and brain abnormalities, and

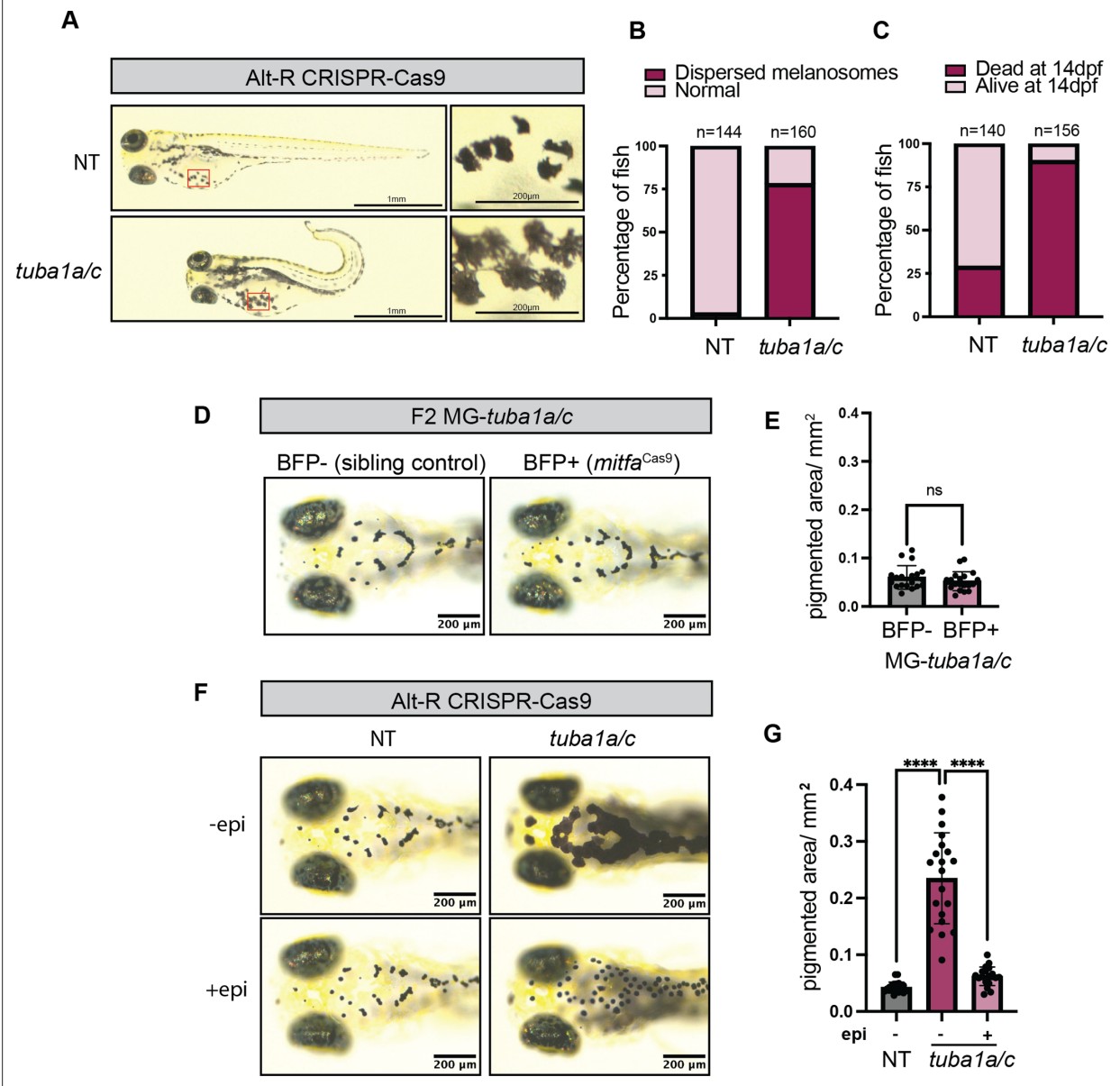

**Figure 4.** Non-autonomous function of *tuba1a/tuba1c* on melanocytes. (**A**) 4 days post-fertilization (dpf) zebrafish embryos injected with either NT or *tuba1a/c* Alt-R CRISPR Cas9 gRNAs. (**B**) Percentage of NT or *tuba1a/c* Alt-R zebrafish embryos with dispersed melanocyte phenotypes. N=2 independent experiments. (**C**) Survival percentage is shown for NT and *tuba1a/c* Alt-R embryos. Embryos were counted at 24 hr post-fertilization (hpf) and again at 14 dpf to determine survival. (**D**) 4 dpf *mitfa*^Cas9 MG-*tuba1a/c* F2 embryos (BFP+ eyes) compared to sibling controls with no *mitfa*^Cas9 (BFP- eyes). Representative embryos are shown for each genotype. (**E**) Pigmented area/mm$^2$ calculated for n=19 embryos/genotype from two independent MG-*tuba1a/c* F2 clutches. Two-sided Student's t-test was used to assess statistical significance, ns: no significance. (**F**) 4 dpf Alt-R-injected zebrafish embryos imaged before and after epinephrine (epi) treatment. (**G**) Pigmented area/mm$^2$ calculated for n=20 embryos/genotype from two independent clutches. Two-sided Student's t-test was used to assess statistical significance, ****p<0.0001.

The online version of this article includes the following figure supplement(s) for figure 4:

**Figure supplement 1.** Knockout of *tuba1a/c*.

morpholino-based knockdown of *tuba1a* in zebrafish inhibits CNS development (**Myers et al., 2015**; **Veldman et al., 2010**).

To further test this idea, we treated *tuba1a/c* Alt-R embryos with epinephrine to assess their ability to contract melanosomes in response to a nonvisual stimulus. *Tuba1a/c* Alt-R embryos exposed to epinephrine had robust aggregation of melanosomes, demonstrating that the loss of *tuba1a/c* did not impair melanosome transport (**Figure 4F and G**). This finding further supports a non-autonomous

role for *tuba1a/c* in melanocytes. Utilizing the *mitfa*<sup>Cas9</sup> system to distinguish cell-autonomous from non-autonomous gene functions provides valuable insights into the complex dynamics of cell-cell interactions and how genes function within cellular and organismal networks.

## Targeting tumor suppressors with *mitfa*<sup>Cas9</sup> induces melanoma

We next turned our attention to melanoma. In humans, tumor suppressors such as *PTEN* and *TP53* are somatically inactivated in melanoma (*Roh et al., 2016*; *Palmieri et al., 2015*). In contrast, the most commonly used zebrafish models of melanoma use a germline *tp53* mutation, which is not typically seen in humans with the disease and precludes our ability to discern the role of *tp53* inactivation specifically in melanocytes and melanoma (*Frantz and Ceol, 2020*). While global *tp53* loss is not lethal, it does lead to a wide range of non-melanoma tumors, which can deteriorate fish health and confound melanoma studies (*Ignatius et al., 2018*). Furthermore, current zebrafish melanoma models typically involve the utilization of MiniCoopR *mitfa* rescue cassettes into *casper* or *nacre* zebrafish, which are devoid of normal pigment cells. Although this method is highly efficient, it does not accurately represent melanoma initiation in the context of normal skin architecture and relies on artificial overexpression of *mitfa*. To mimic human melanoma initiation more accurately, we used our *mitfa*<sup>Cas9</sup> model to assess tumor initiation using melanoma-specific gene knockout in animals with WT skin. Previous studies have shown that expression of BRAF<sup>V600E</sup> and loss of *tp53* in WT (non-*casper*) fish results in relatively low tumor burden, suggesting that the casper strain might be especially tumor prone in the setting of MiniCoopR *mitfa* rescue, although the exact reasons for this are not clear. Based on this, to increase penetrance, we decided to also target *ptena* and *ptenb*, the zebrafish orthologs of human *PTEN* (*Patton et al., 2005*). *Ptena/ptenb* loss in combination with *tp53* has been shown to accelerate melanoma formation in zebrafish (*He et al., 2021*; *Montal et al., 2024*).

To first address the effect of *ptena/b* loss on normal melanocytes, we generated melanocyte-specific knockouts of *ptena* and *ptenb* (*Figure 5A*). To visualize cells expressing both *ptena* and *ptenb* guides, we designed additional plasmids, denoted MTdT-gRNA, containing a zU6:gRNA cassette followed by *mitfa*:TdTomato and co-injected MG-*ptena* and MTdT-*ptenb* plasmids into *mitfa*<sup>Cas9</sup> embryos. Like *sox10* and *tuba1a/c*, germline knockout of *ptena/b* is embryonic lethal (*Croushore et al., 2005*; *Faucherre et al., 2008*). In our *ptena/ptenb* F0 melanocyte-specific KO fish, survival was normal. We observed an aberrant expansion of the melanocytes outside of the stripe regions they normally are restrained to, suggesting that inactivation of *ptena* and *ptenb* induces defects in melanocyte patterning (*Figure 5A and B*). However, loss of *ptena/b* alone was not sufficient to induce melanoma.

To generate tumors, we co-injected *mitfa*<sup>Cas9</sup> embryos with plasmids encoding *mitfa*:BRAF<sup>V600E</sup> and gRNAs targeting *tp53* and *ptena/b* (*Figure 5C*). Larvae were sorted for BFP+ eyes, indicating mitfa-Cas9 integration, and monitored for tumors over 30 weeks. No tumors were detected in BFP- fish from any of the injection conditions (n=136). We also did not observe any tumors in BFP+ fish injected with either BRAF<sup>V600E</sup> alone (n=35) or ptena/b; *tp53* guides in the absence of BRAF<sup>V600E</sup> (n=17). In contrast, tumors developed in all BFP+ groups injected with both *mitfa*:BRAF<sup>V600E</sup> and gRNA plasmids. Approximately 5% of fish from the BRAF;*tp53* group developed tumors (*Figure 5D*). This aligns with a previous study where 6% of *tp53*-/- fish developed tumors when injected with a BRAF<sup>V600E</sup> construct (*Patton et al., 2005*). Targeting of *ptena* and *ptenb* in *mitfa*<sup>Cas9</sup> fish co-injected with *mitfa*:BRAF<sup>V600E</sup> resulted in a higher tumor incidence of 29% (*Figure 5D*).

When all three tumor suppressors (*tp53*, *ptena*, *ptenb*) were targeted in combination with BRAF<sup>V600E</sup>, 59% of fish developed tumors by 30 weeks post-fertilization (wpf) (*Figure 5D*). Immunohistochemistry (IHC) revealed that all tumor samples expressed BRAF<sup>V600E</sup>, while p-AKT, a hallmark of *PTEN* inactivation, was only detected in samples with *ptena/b* KO (*Figure 5E*). To confirm on-target cutting, DNA was extracted from tumors, and CRISPR sequencing was performed on PCR amplicons for each tumor suppressor gene. A large proportion of reads contained indels at the target locus for each of the target genes (*ptena*, *ptenb*, and *tp53*) (*Figure 5F*, *Figure 5—figure supplement 1A–C*). Some tumors had an array of mutations, while others appeared to be composed largely of a one dominant clone (*Figure 5—figure supplement 1A–C*). Thus, similar to the MAZERATI system, our endogenous knock-in system allows us to rapidly dissect the melanocyte-specific function of both oncogenes (BRAF<sup>V600E</sup>) and putative tumor suppressors (*tp53*, *ptena/b*), while avoiding pleiotropic effects of global tumor suppressor loss.

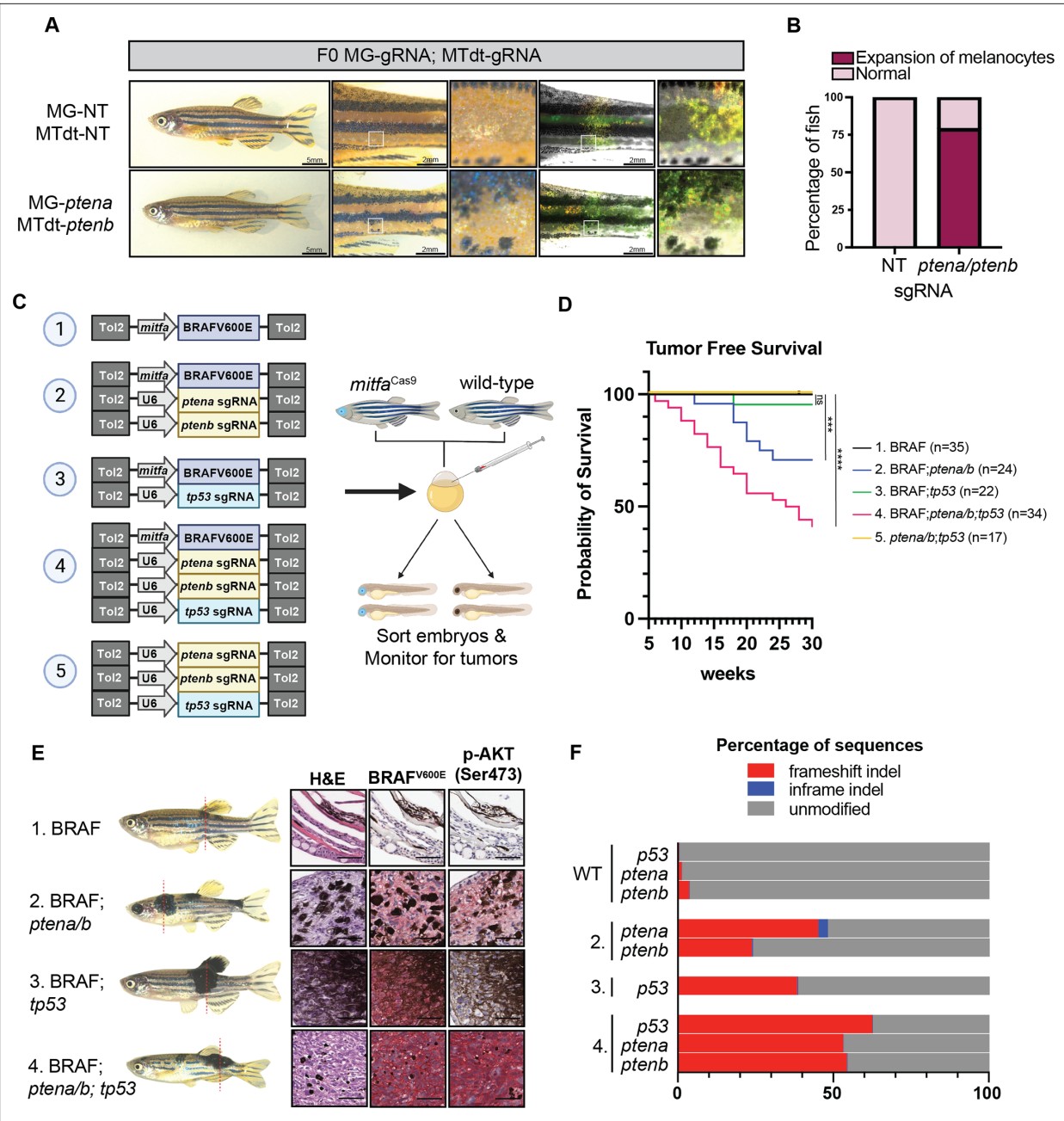

**Figure 5.** Generation of a zebrafish melanoma model in *mitfa*Cas9 fish. (**A**) Adult *mitfa*Cas9 MG-NT; *mitfa*:TdTomato;U6:NT (MTdt-NT) and MG-*ptena*; *mitfa*:TdTomato;U6:*ptenb* (MTdT-*ptenb*) F0 fish. (**B**) Proportion of MG-NT; MTdT-NT (n=14) and MG-*ptena*; MTdT-*ptenb* (n=14) F0 fish with disrupted stripes phenotype. (**C**) Schematic of zebrafish tumorigenesis assay. Indicated plasmids are injected into one-cell-stage embryos from crosses between *mitfa*Cas9 and wild-type (WT) fish. The *mitfa*:BRAFV600E plasmid includes cardiac-specific cmlc2:GFP. Embryos were sorted for GFP+ hearts and BFP+ eyes, and fish were screened every 2 weeks for tumors. Created with BioRender.com. (**D**) Tumor-free survival curve. N=2 independent experiments. Tumors were tracked over the course of 30 weeks. Log-rank (Mantel-Cox) test was used to assess statistical significance, ***p<0.001; ****p<0.0001; ns: no significance. (**E**) Histology was performed on one fish from each indicated injection group. Dotted lines indicate the site of sectioning. Red chromogen was used for all immunohistochemical (IHC) staining. Scale bars, 50 μm. (**F**) CRISPR-seq results are shown for normal skin dissected from WT fish and tumors dissected from injection conditions 2, 3, and 4. Results are shown as a fraction of sequences with indels calculated using CRISPResso.

The online version of this article includes the following figure supplement(s) for figure 5:

**Figure supplement 1.** CRISPR-seq for tumor suppressor genes.

## Targeting lineage-specific oncogenes decreases tumor initiation but promotes progression

Our next objective was to leverage the *mitfa*<sup>Cas9</sup> fish as a tool for identifying or validating potential genetic dependencies within melanoma. Genetic dependencies can be difficult to study in vivo since inactivation of a required gene inevitably leads to selection for non-mutant alleles, as has been previously shown (*Kaufman et al., 2016*). Large-scale in vitro efforts such as the DepMap can provide guidance as to what genes are likely to be important for tumor proliferation, but cell line studies alone cannot recapitulate the complex in vivo behavior that occurs from knocking individual genes out. In melanoma, the DepMap has identified *SOX10* as the top in vitro genetic dependency necessary for tumor cell proliferation, suggesting it acts as a lineage-specific oncogene (*Tsherniak et al., 2017*; *Shakhova et al., 2015*). In prior work in both zebrafish and mice, loss of *sox10* is clearly associated with a loss of tumor-initiating potential and cell proliferation (*Kaufman et al., 2016*; *Shakhova et al., 2015*). Yet its specific role in vivo remains unclear, since it is required for neural crest and melanocyte specification, making it difficult to know whether its effects reflect loss of the entire lineage (*Shakhova et al., 2015*). Given our data above showing that *sox10* is required specifically in melanocyte patterning and regeneration, we wished to investigate its function in melanoma.

To assess this, we generated plasmids containing multiple gRNAs driven by three distinct zU6 promoters (zU6A, zU6B, and zU6C) to simultaneously target both tumor suppressors (*tp53, ptena/ptenb*) and tumor-promoting genes (*sox10*) in the same cells, which reduces the chance of selecting for non-mutant alleles (*Yin et al., 2015*). We co-injected these plasmids with *mitfa*:BRAF<sup>V600E</sup>, sorted fish for BFP+ eyes, and tracked tumor-free survival over the course of 50 weeks (*Figure 6A and B*). Consistent with the DepMap prediction, *sox10* inactivation markedly reduced tumor incidence, which likely reflects loss of proliferation. Only 3/96 (3.1%) fish with *sox10* knockouts initiated melanomas compared to the 24/94 fish (25.5%) from the NT condition that developed tumors (*Figure 6B*). Using IHC, we validated that *sox10* KO tumors expressed lower levels of Sox10 protein compared to NT tumors (*Figure 6C*, *Figure 6—figure supplement 1A*). Quantification of Sox10 intensity with immunofluorescence confirmed this finding (*Figure 6—figure supplement 1B*).

Although *sox10* clearly reduced tumor initiation, we noted 'escapers'. These tumors appeared morphologically somewhat distinct compared to the *sox10* intact tumors. While not easily quantifiable, qualitatively based on prior experience, the tumor cells appeared more mesenchymal and penetrated more deeply under the skin than we typically see with the BRAF<sup>V600E</sup> melanomas in the fish (*Figure 6C*). This observation raised the possibility that these tumors had undergone 'phenotype switching', a form of cell plasticity in which cells reversibly move between opposite extremes of proliferation versus invasive states (*Capparelli et al., 2022*; *Wouters et al., 2020*; *Shakhova et al., 2015*; *Hoek et al., 2008*). In melanoma, the proliferative state is thought to be characterized by high expression of *SOX10*, whereas the mesenchymal, invasive state is characterized by high expression of *SOX9* (*Wouters et al., 2020*). These two highly related transcription factors have seemingly opposite and possibly antagonistic roles in melanoma. In vitro, knockout of *SOX10* in a variety of human cell lines is associated with acquisition of a *SOX9*<sup>hi</sup> state, which has been suggested to be linked to the invasive phenotype (*Wouters et al., 2020*). We performed a re-analysis of publicly available RNA-seq data using three patient-derived human melanoma cell lines treated with siRNA targeting *SOX10* confirmed this observation, with the *SOX10* lines now becoming *SOX9*<sup>hi</sup> (*Figure 6E–J*, *Figure 6—figure supplement 1C–L*; *Wouters et al., 2020*). These data raised the possibility that the more mesenchymal, invasive-appearing tumors we see in our escaper fish might be due to gain of *sox9* expression. To test this, we stained our *sox10* CRISPR tumors for *sox9* and discovered that two of the three *sox10* KO tumors markedly upregulated *sox9* relative to control tumors expressing NT gRNA (*Figure 6C and D*). While this data is certainly not definitive, when taken in context of the above in vitro and in vivo data in other model systems, it appears consistent with the idea that loss of *sox10* could result in more invasive tumors. This preliminary observation we have made will need further mechanistic dissection, and our system would be amenable to studying this phenomenon in further depth. This data also highlights that in vitro genetic studies such as the DepMap, which rely on proliferation as the readout, can mask more complex in vivo phenotypes.

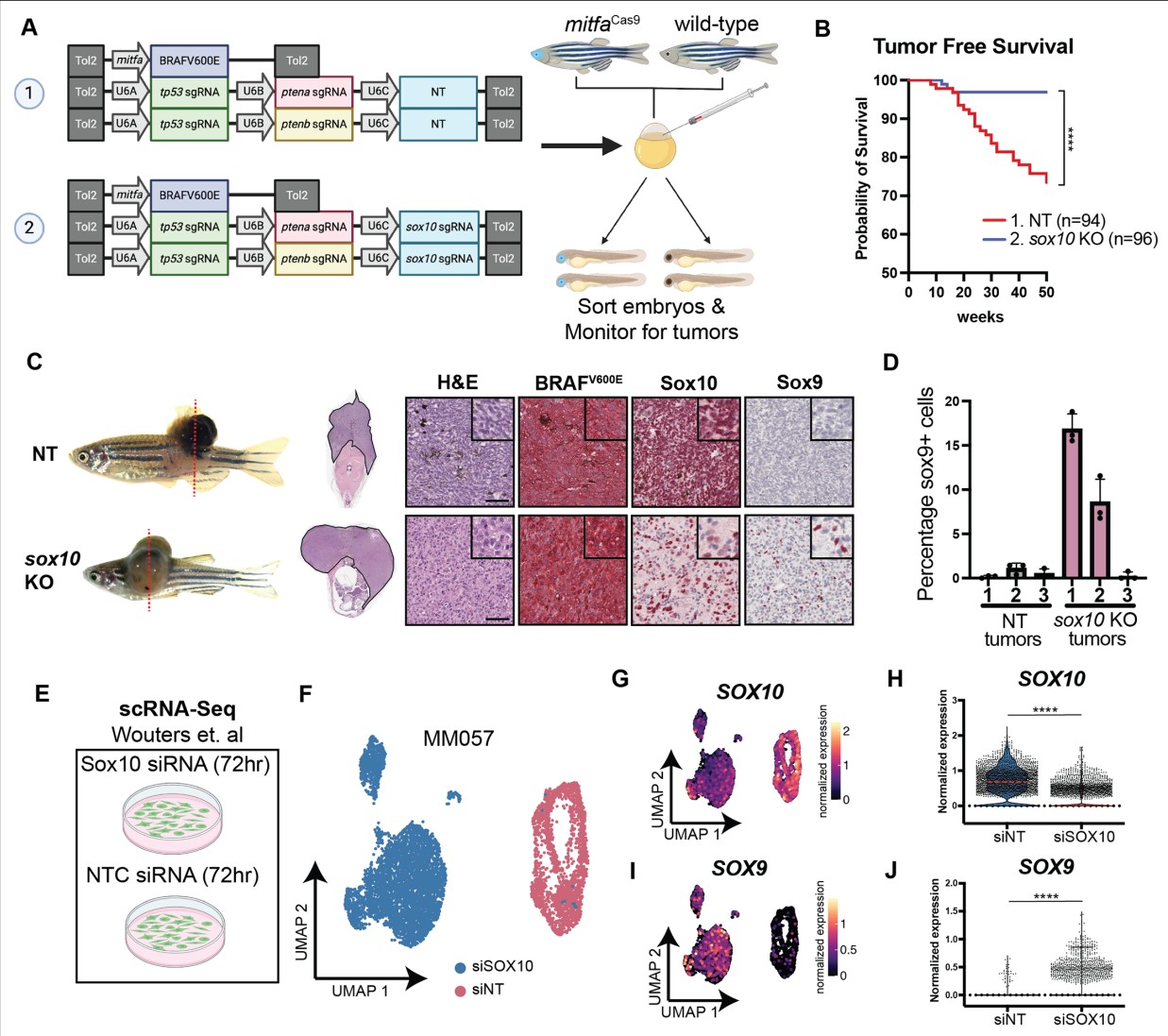

**Figure 6.** Melanoma-specific knockout of *sox10* reduces tumor burden and induces phenotypic switching. (**A**) Schematic of zebrafish tumorigenesis assay. Indicated plasmids are injected into one-cell-stage embryos from crosses between *mitfa*^Cas9 and wild-type (WT) fish. Embryos were sorted for GFP+ hearts and BFP+ eyes, and fish were screened every 2 weeks for tumors. Created with BioRender.com. (**B**) Tumor-free survival curve. N=3 independent experiments. Tumors were tracked over the course of 50 weeks. Log-rank (Mantel-Cox) test was used to assess statistical significance, ****p<0.0001. (**C**) Histology is shown for one fish from each injection group. Dotted lines indicate the site of sectioning. Red chromogen was used for all immunohistochemical (IHC) staining. Scale bars, 50 μm. (**D**) Color thresholding was used on IHC images to calculate the percentage of nuclei that stained positive for *sox9*. NT-1 n=1222 cells, NT-2 n=1620 cells, NT-3 n=2185 cells, *sox10-1* n=970 cells, *sox10*-2 n=594, *sox10*-3 n=1046. Cells from n=3 images were analyzed for each tumor. (**E**) Schematic of scRNA-seq experimental setup from Wouters et al. Patient-derived cell lines were treated with siRNAs targeting SOX10 or NTC, and scRNA-seq was conducted at 72 hr. (**F**) UMAP of scRNA-seq dataset for MM057 cell line from Wouters et al. Cells from siSOX10 and siNT conditions are labeled. (**G**) Normalized expression of Sox10 per cell in UMAP space. (**H**) Violin plots of normalized expression of Sox10 per cell. Median is shown as dashed red line. Wilcoxon rank sum test was used to assess statistical significance, ****p<0.0001. (**I**) Normalized expression of Sox9 per cell in UMAP space. (**J**) Violin plots of normalized expression of Sox9 per cell. Wilcoxon rank sum test was used to assess statistical significance, ****p<0.0001.

The online version of this article includes the following figure supplement(s) for figure 6:

**Figure supplement 1.** In vivo and in vitro targeting of Sox10 leads to upregulation of Sox9.

## Discussion

The ability to model genetic dependencies in vivo is one of the most important uses of model organisms. Forward genetic mutagenesis screens using compounds such as ENU or EMS led to the identification of many key developmental genes in organisms such as *Drosophila*, *Caenorhabditis elegans*,

*Saccharomyces*, zebrafish, and many others (**Solnica-Krezel et al., 1994**; **Russell et al., 1979**). These approaches have rapidly accelerated in the era of TALENs and CRISPR, readily allowing for knockout of nearly any gene in a wide variety of non-model organisms. Cre/Lox approaches have further augmented our ability to discern the cell-type-specific effects of these genes, although these are still laborious and time-consuming, especially in models such as mice (**Shin et al., 2023**).

Alongside these approaches have been remarkable advances in large-scale sequencing of human tissues. GWAS-like efforts such as the UK Biobank and the US All of Us programs have identified thousands of germline variants (SNPs) that may be linked to interesting phenotype variation (**Mayer and Huser, 2023**; **Sudlow et al., 2015**). A conceptually similar approach is also happening in diseases such as cancer, in which efforts such as the TCGA, ICGC, and DepMap have continued to unveil new potential somatic variants (SNVs) linked to cancer (**Tsherniak et al., 2017**; **Cerami et al., 2012**; **Alexandrov et al., 2013**).

Across these efforts, in both development and disease, the large number of candidate variants makes it challenging to connect these genotypes to phenotypes. This is especially acute in melanoma, where the large number of somatic variants generated by UV radiation leads to a high number of background mutations that may or may not have pathogenic function. Thus, there is a substantial need for in vivo models that allow for rapid and efficient modeling of candidate genes.

Prior work in the zebrafish has shown that promoter fragments driving Cas9 can provide efficient and scalable approaches to this problem (**Ablain et al., 2015**; **Ablain et al., 2021**). For example, MAZERATI has been nicely applied to melanoma models to study genetic dependencies in the melanocyte lineage (**Ablain et al., 2021**). The technology we describe in this work largely builds off the logic of this and related systems. One difference between our *mitfa*-Cas9 knock-in line compared to MAZERATI is that it maintains the endogenous regulatory elements needed for control of the *mitfa* gene. We do not yet fully know when or if this difference would be important, but given the complex role of MITF transcription as a 'rheostat' in melanoma (**Hoek and Goding, 2010**), it is reasonable to assume these endogenous elements could become important in some circumstances. Because in our study we did not directly compare the *mitfa*-Cas9 knock-in approach versus MAZERATI, we cannot make direct statements about which is more efficient. However, because both techniques reliably result in melanoma, each researcher will be able to choose the technology that is most appropriate for their biological question.

One advantage of Cas9 approaches in this system is the ability to screen phenotypes in both the F0 generation (as mosaics) and the F1 generation (3 months later). Although the F0 phenotypes we observe are indeed subtle, they are still quantifiable. This enables rapid screening of candidate genes for melanocyte-specific functions. Promising hits can then be more specifically validated in the F1 generation, which is particularly advantageous for efficiently generating biallelic knockouts, especially for genes with recessive phenotypes like *albino*. On a per-gene basis, this saves well over 3 months of work compared to germline recessive knockouts and eliminates the issue of inbreeding, promoting healthier genetic lines and facilitating the potential for outcrossing with any available fish line. Because we demonstrate that Cas9 is completely restricted to the melanocyte lineage, with no detectable off-target expression, this easily allows us to discern the melanocyte-specific effect of pleiotropic genes very efficiently.

One interesting observation of our melanoma studies is evidence that loss of melanoma genetic dependencies like *SOX10* (as suggested by the human DepMap data) can still lead to a small number of tumors, albeit with seemingly different biological behaviors. In the human samples we analyzed, loss of *SOX10* leads to acquisition of a *SOX9*[hi] state instead (**Wouters et al., 2020**). Similarly, in our 'escaper' fish, we do see an increase in SOX9 expression in two out of three fish, which would be consistent with the human data. *SOX9* is increasingly recognized to be a potent mediator of tumor cell invasiveness, so it is possible this could explain the somewhat more mesenchymal/invasive appearance of these tumors (**Cheng et al., 2015**; **Yang et al., 2019**). This phenomenon clearly needs further study in future work, where the effect could be better quantified in a larger cohort and the relevant mechanism identified. But if this SOX10 to SOX9 switch holds up, one implication of this finding is that clinical targeting of transcription factors like *SOX10* (**Takahashi et al., 2024**) – long considered an ideal therapeutic approach – could lead to unexpected outcomes, in which tumors might proliferate less but become more invasive or resistant to therapy. Whether this phenomenon of *SOX10/SOX9* switching is a general phenomenon, or specific to that class of TFs, remains to be determined.

However, a similar phenomenon has been observed for *MITF*, another melanocytic TF, suggesting this idea of TF switching needs to be more thoroughly explored in the future (*Hoek et al., 2008*; *Hossain and Eccles, 2023*; *Travnickova et al., 2019*).

Finally, while our studies clearly focused on the melanocyte lineage, highly similar approaches could be taken for virtually any cellular lineage in which master regulators or markers are known. Given the large number of already identified promoter/enhancer fragments in the zebrafish, this would be an important extension of our system to study gene variants in the context of development and other diseases (*Weiss et al., 2022b*).

Our approach has several limitations that may need to be addressed in future studies. Due to the constitutive nature of the Cas9, we have little control over the timing of its expression. Because it is knocked into the endogenous *mitfa* locus, the expression of Cas9 will vary with the endogenous regulation of that gene. This may lead to somewhat paradoxical and unexpected results. For example, germline deletion of *sox10* results in the colorless phenotype, which is an animal completely devoid of melanocytes (*Dutton et al., 2001*). In contrast, when we knock out *sox10* specifically in the *mitfa+* cells with our *mitfa*Cas9 system, we see a much more subtle phenotype, and not all melanocytes are lost. We speculate this could be due to timing: the germline allele will already have loss of *sox10* function very early in the development of the melanocyte lineage (i.e. at the neural crest stage), whereas our knock-in approach will inactivate *sox10* much later in development after melanocyte specification has already occurred. This could reflect different downstream targets of *sox10* in these two scenarios. In future iterations of this general approach, swapping the Cas9 for an inducible version would allow for better-timed knockout of genes (*Sun et al., 2019*). Because *mitfa* itself is expressed in melanoblasts, melanophores, and xanthophores during different stages of development, choosing a promoter (e.g. PMEL) that is more tightly restricted to melanocytes may be advantageous. Another limitation is that our fish will result in a single functional copy of *mitfa*. While this does not result in any obvious phenotype (i.e. the melanocytes are normal) and this gene is not currently known to exhibit haploinsufficiency, it is possible that under certain circumstances, the loss of one copy of *mitfa* could have unintended phenotypes. For example, in the tumor setting, having only one copy of *mitfa* could theoretically reduce tumor incidence since it can act as an oncogene in some situations (*Garraway et al., 2005*). This possibility of haploinsufficiency will need to be further explored in the future. Finally, although our system was relatively efficient, we still did not see 100% knockout. This may reflect that it is relatively easy to bypass the CRISPR lesion in an exon with an in-frame mutation. The use of multiple gRNAs, or gRNAs that target promoter regions, rather than exons, could further augment efficiency of knockout. Alternatively, instead of targeting DNA, we could consider using RNA-targeting CRISPR enzymes. Recent work has shown that RNA degradation induced by Cas13 works in zebrafish and may allow for more nuanced knockdown rather than knockout of a given candidate gene (*Huang et al., 2023*; *Kushawah et al., 2020*). This may especially have advantages in the context of melanoma, where genes are commonly dysregulated rather than completely knocked out. We envision it would be relatively straightforward to knock Cas13 into the locus of interest using an analogous approach to the one we have taken here.

## Methods
### Materials availability
Plasmids, fish lines, and other materials generated from this study are available upon request.

### Zebrafish husbandry and ethics statement
All zebrafish experiments adhered to institutional animal protocols and were conducted in compliance with approved procedures. Fish stocks were maintained at a temperature of 28.5°C, under 14:10 light:dark cycles, with pH set at 7.4, and salinity-controlled conditions. The zebrafish were fed a standard diet comprising brine shrimp followed by Zeigler pellets. Approval for the animal protocols outlined in this manuscript was obtained from the Memorial Sloan Kettering Cancer Center (MSKCC) Institutional Animal Care and Use Committee (IACUC), under protocol number 12-05-008. Anesthesia was conducted using Tricaine (4 g/l, Syndel, Ferndale, WA, USA) from a stock of 4 g/l and diluted to 0.16 mg/ml. Adult zebrafish of both sexes were equally employed in all experiments. Embryos,

collected through natural mating, were incubated in E3 buffer (5 mM NaCl, 0.17 mM KCl, 0.33 mM CaCl$_2$, 0.33 mM MgSO$_4$) at 28.5°C. The WT strain used was T5D zebrafish (*Balik-Meisner et al., 2018*).

## GeneWeld plasmid construction

A combination of restriction enzyme digestion and HiFi cloning was used to construct the GeneWeld pPRISM-nCas9n, γcry1:BFP vector for targeted integration to isolate an nls-Cas9-nls knock-in. A 765 bp fragment containing the Porcine teschovirus-1 polyprotein 2A peptide sequence was amplified from the pPRISM-2A-Cre, γcry1:BFP vector (Addgene #117789) with primers Ori-F and 2 A-R (*Welker et al., 2021*, *Wierson et al., 2020*). The nCas9n cDNA was amplified from the expression vector pT3TS-nCas9n (Addgene #46757) with primers Cas9-F and Cas9-R (*Jao et al., 2013*). A KpnI/SpeI fragment containing the pPRISM-2A-Cre, gcry1:BFP vector backbone was assembled with the 2A and nCas9n PCR amplicons using the NEBuilder HiFi DNA Assembly Cloning Kit (NEB # E5520S) following the manufacturer's instructions.

Homology arms were designed as previously described (*Welker et al., 2021*; *Wierson et al., 2020*). Briefly, 48 bp homology arms complementary to the *mitfa* target site were designed using GTagHD (http://www.genesculpt.org/gtaghd/) for the pPRISM GeneWeld plasmid series. The two pairs of complementary oligos can be found in *Supplementary file 1*. 5' and 3' arms were cloned sequentially into the pPRISM-nCas9n, γcry1:BFP vector. The 3' homology arm oligos were annealed and cloned into the vector using the BspQI restriction site as previously described. Due to the multiple BfuAI sites within Cas9, instead of using BfuAI restriction cloning, we alternatively used in-fusion cloning (Takara Bio) to insert a gBlock (IDT) containing the 5' homology arm into the pPRISM-nCas9n, gcry1:BFP vector. Primers and gblock sequences used for cloning are listed in *Supplementary file 1*. The 5' and 3' homology arms were sequenced with primers 5'homologycheckF and R3'_pgtag_seq, respectively.

## Injection of GeneWeld reagents

Injections of GeneWeld reagents were performed on one-cell-stage embryos. We used the Alt-R CRISPR-Cas9 system (IDT) for the initial knock-in of nCas9n, γcry1:BFP to the *mitfa* locus. In brief, 100 µM tracrRNA (IDT 1075928) was mixed at a 1:1 ratio with either *mitfa* crRNA or universal crRNA, then incubated at 95°C for 5 min. Recombinant Cas9 (IDT 1081059) was incubated with both gRNAs for 10 min at 37°C to form the RNP complex. The GeneWeld pPRISM-5'mitfaHom-nCas9n, gcry1:BFP-3'mitfaHom plasmid was then added to the injection mix along with phenol red (Sigma-Aldrich P0290). The final concentrations injected into the embryos were 75 pg/nl Cas9, 12.5 pg/nl *mitfa* gRNA, 12.5 pg/nl Universal gRNA, and 5 pg/nl pPRISM plasmid.

## Genotyping

Tail clips were taken from 2 month post-fertilization (mpf) zebrafish, and DNA was extracted using the DNeasy Blood and Tissue Kit (QIAGEN). DNA was PCR-amplified with Q5 high-fidelity DNA polymerase (NEB). The forward primer MitfaPCR-F binds to the *mitfa* genomic region upstream of the insertion site and the reverse primer 5'homology_amplify-R binds the p2A region of the insertion cassette. Expected PCR amplicon size is 269 bp. Amplicons were sequenced using Sanger sequencing (Azenta Life Sciences).

## RNA-FISH HCR

We adapted the HCR protocol from *Ibarra-García-Padilla et al., 2021*. Briefly, embryos were collected and treated with 0.0045% 1-phenyl 2-thiourea (Sigma-Aldrich) after 24 hpf to block pigmentation. Embryos were harvested at 72 hpf and fixed in 4% PFA (Santa Cruz) for 24 hr at 4°C. Embryos were then washed with PBS and dehydrated/permeabilized with a series of MeOH (Millipore) washes. Embryos were rehydrated with a series of MeOH/PBST washes, permeabilized with acetone (Fisher Scientific), and incubated with Proteinase K (Millipore) for 30 min. This was followed by further fixation with 4% PFA. Embryos were pre-hybridized in probe hybridization buffer (PHB) before incubating them with HCR probes in PHB at a concentration of 20 nM. Probe sequences for *mitfa* and Cas9 can be found in *Supplementary file 1*. Embryos were washed with probe wash buffer before being pre-amplified with probe amplification buffer (PAB). Following pre-amplification, amplifier hairpins were thawed, snap-cooled, and mixed with PAB to a concentration of 36 nM. Embryos were incubated in

the hairpin solution followed by washes with 5 × SSCT and 1 × PBST. Embryos were stained with 2 μg/ml DAPI in PBST and mounted on glass slides. Imaging was performed using a Zeiss LSM880 confocal microscope. Melanocytes were assessed for co-expression of *mitfa* and Cas9 within a 212-μm-long section of the embryo, located directly caudal to the yolk, and typically encompassing three to five distinct midline melanocytes per embryo. Overlap in expression was assessed using Fiji.

## Guide RNA plasmids

The following plasmids were constructed using the Gateway Tol2kit:

zU6A:gRNA-NT;zU6B:gRNA-*ptena*;zU6C:gRNA-*tp53*/394,
zU6A:gRNA-NT;zU6B:gRNA-*ptenb*;zU6C:gRNA-*tp53*/394,
zU6A:gRNA-*sox10*;zU6B:gRNA-*ptena*;zU6C:gRNA-*tp53*/394,
zU6A:gRNA-*sox10*;zU6B:gRNA-*ptenb*;zU6C:gRNA-*tp53*/394.

In-fusion cloning (Takara Bio) was used to insert *mitfa:*GFP or *mitfa:*TdTomato into a zU6:gRNA/394 plasmid previously developed in the lab (see **Supplementary file 1** for primers used). gRNAs were cloned into zU6:gRNA;*mitfa:*GFP/394 or zU6:gRNA;*mitfa:*TdTomato/394 using BsmBI sites. Other plasmids used in this study that were previously developed in the lab include zU6:gRNA-*ptena*/394, zU6:gRNA-*ptenb*/394, zU6:gRNA-*tp53*/394, *mitfa*-BRAF$^{V600E}$;cmlc2:eGFP. All guide RNAs used in this study besides *tuba1a/c* have previously been validated in zebrafish (**Welker et al., 2021**; **Weiss et al., 2022a**; **Kaufman et al., 2016**; **Suresh et al., 2023**).

## Validation of *tuba1a/c* gRNAs

The Alt-R CRISPR-Cas9 system (IDT) was used to achieve non-cell-type specific knockout of *tuba1a/c*. Guide RNAs were designed using ChopChop (**Labun et al., 2019**). In brief, 100 μM tracrRNA (IDT 1075928) was mixed at a 1:1 ratio with crRNA, then incubated at 95°C for 5 min. Recombinant Cas9 (IDT 1081059) was incubated with the gRNA for 10 min at 37°C to form the RNP complex. Phenol red was added prior to injection into one-cell-stage WT T5D embryos. To validate on-target cutting, four embryos were pooled, and DNA was extracted using Quick Extract buffer (Fisher Scientific NC0302740). DNA was PCR-amplified with Q5 high-fidelity DNA polymerase (NEB) and forward and reverse primers (**Supplementary file 1**). Exosap IT for PCR Product Clean Up was added to PCR product prior to Sanger sequencing. Forward and reverse primers used for PCR amplification were also used for sequencing (**Supplementary file 1**).

## Transgenic lines

To generate MG-U6:gRNA stable lines, one-cell-stage embryos from crossing WT and mitfa:Cas9 fish were collected and injected with the 30 pg indicated plasmid and 20 pg tol2 mRNA. Fish with GFP+ melanocytes and BFP+ eyes were selected and outcrossed with WT fish to produce the F1 generation. F1 fish were again sorted and outcrossed to generate F2 generation zebrafish. All fish were imaged on a Zeiss AxioZoom V16 with Zen 2.1 software.

## Flow cytometry of melanocytes from adult zebrafish

Zebrafish were euthanized using ice-cold water and dissected using a clean scalpel and forceps. The epidermal and dermal layers of the skin, as well as the fins, were separated from the rest of the tissues and diced into 1–3 mm pieces. Cells were dissociated with Liberase TL (Millipore Sigma 05401020001) and filtered for single-cell suspensions as previously described (**Weiss et al., 2022a**). Samples were then FACS-sorted (BD FACSAria) for GFP+ and GFP- cells. WT fish were used as a GFP-negative control. Genomic DNA was isolated from sorted cells using the DNeasy Blood and Tissue Kit (QIAGEN).

## Tumor dissection

Fish with tumors were euthanized with ice-cold water and immediately dissected with a scalpel to isolate tumor tissue. Genomic DNA was purified from the tumor samples using the DNeasy Blood and Tissue Kit (QIAGEN).

## CRISPR sequencing

Primer pairs were designed to produce amplicons 200–280 bp in length with the mutation site within 100 bp from the beginning or end of amplicon (see *Supplementary file 1* for primer sequences). Genomic DNA isolated from methods detailed above was PCR-amplified with Q5 high-fidelity DNA polymerase (NEB) and run on an agarose gel to visualize PCR products. The samples were then purified using the NucleoSpin Gel and PCR Cleanup Kit (Takara Bio). Deep sequencing was conducted using the CRISPR-seq platform. Sequencing data was analyzed with CRISPResso2 (*Clement et al., 2019*), and indel charts were generated with CRISPRVariants R package (*Lindsay et al., 2016*).

## Drug treatments

Adult zebrafish were treated for 24 hr with 750 nM neocuproine (Sigma-Aldrich) dissolved in fish water with a final concentration of 0.0075% DMSO as previously described (*O'Reilly-Pol and Johnson, 2008*). Cells from the middle melanocyte stripe were counted within a rectangular region of 4 mm × 1.5 mm. Epinephrine hydrochloride (Sigma-Aldrich) was used at 1 mg/ml in fish water. Fish were treated for 10 min prior to imaging.

## Zebrafish tumor-free survival curves

To generate a transgenic melanoma model, one-cell-stage embryos from WT X *mitfa*:Cas9 fish crosses were injected with *mitfa*-BRAF$^{V600E}$;cmlc2:eGFP, 20 pg Tol2 mRNA, and the indicated gRNA plasmids for each condition. The total amount of plasmid per embryo did not exceed 30 pg. Embryos were screened for GFP-positive hearts and sorted for BFP+ or BFP- eyes. Fish were screened for tumors using a microscope every 2 weeks starting at 4 wpf. Tumors were only called if an elevated lesion was observed. GraphPad Prism 9 was used to generate and analyze Kaplan-Meier survival curves. Statistical differences were determined using the log-rank Mantel-Cox test. All screening and imaging was conducted on a Zeiss AxioZoom V16 using Zen 2.1 software.

## Histology

Zebrafish were sacrificed using ice-cold water and placed in 4% paraformaldehyde (Santa Cruz 30525-89-4) in PBS for 72 hr at 4°C on a rocker. Fish were then transferred to 70% EtOH for 24 hr at 4°C on a rocker. Fish were sent to Histowiz, Inc (Brooklyn, NY, USA), where they were paraffin-embedded, sectioned, stained, and imaged. All the stainings were performed at Histowiz, Inc Brooklyn, using the Leica Bond RX automated stainer (Leica Microsystems) using a Standard Operating Procedure and fully automated workflow. Samples were processed, embedded in paraffin, and sectioned at 4 µm. The slides were dewaxed using xylene and alcohol-based dewaxing solutions. Epitope retrieval was performed by heat-induced epitope retrieval of the formalin-fixed, paraffin-embedded tissue using citrate-based pH 6 solution (Leica Microsystems, AR9961) for 20 min at 95°C. The tissues were first incubated with peroxide block buffer (Leica Microsystems), followed by incubation with the following antibodies for 30 min: BRAFV600E antibody (ab228461, 1:100), phospho-AKT antibody (CST4060, 1:50), SOX9 antibody (AB_185230, 1:1000), and SOX10 antibody (GTX128374, 1:500). This was followed by incubation with DAB secondary reagents: polymer, DAB refine, and hematoxylin (Bond Polymer Refine Detection Kit, Leica Microsystems) according to the manufacturer's protocol. The slides were dried, coverslipped (TissueTek-Prisma Coverslipper), and visualized using a Leica Aperio AT2 slide scanner (Leica Microsystems) at 40X.

## Quantification of Sox10 intensity

FFPE slides were deparaffinized by consecutive incubations in xylene and 100-50% ethanol. For antigen retrieval, slides were placed in 10 mM sodium citrate pH 6.2 in a pressure cooker and heated to 95°C for 20 min. Slides were then cooled to room temperature and blocked in 5% donkey serum, 1% BSA, and 0.4% Triton X-100 in PBS for 1 hr at room temperature. The primary antibody targeting Sox10 (GeneTex, GTX128374) was added to the blocking buffer at 1:200 concentration and incubated at 4°C overnight. Slides were then washed in PBS before incubation with the secondary antibody (donkey anti-rabbit Alexa Fluor Plus 488, Thermo Fisher Scientific #AC32790; 1:250) and Hoescht (Thermo Fisher Scientific, #62249; 1:1000) in blocking buffer for 2 hr at room temperature. Slides were mounted in Vectashield Plus (Vector Labs, H-1900). Slides were imaged on an LSM 880 confocal microscope using a ×40 oil immersion objective. 3 images were captured per sample. Quantification

of Sox10 intensity per cell was performed using CellProfiler (*Stirling et al., 2021*) and MATLAB r2023b (Mathworks). Cells were segmented in CellProfiler to obtain a nuclear mask from the Hoescht signal, and using this mask, the mean intensity per nucleus was quantified.

## Re-analysis of human melanoma scRNA-seq data

Human melanoma scRNA-seq data from *Wouters et al., 2020* was accessed from GEO (GSE134432). All analysis was done in R (version 4.3.1) using the Seurat package (version 5.0.2) (*Hao et al., 2024*). Counts matrices for each sample and cell line were imported into R using the function read.table and converted into Seurat objects with the function CreateSeuratObject before merging into one combined Seurat object using the function merge. Counts were normalized using the Seurat function NormalizeData with default parameters. UMAP was calculated using the Seurat function RunUMAP with 10 dimensions. SOX10 expression was plotted using the functions FeaturePlot and VlnPlot.

## Statistical analysis

Statistical significance was analyzed using t test, log-rank (Mantel-Cox) test, Wilcoxon rank sum test, or as indicated in the figure legends. GraphPad Prism 9 software was used for data processing and statistical analysis. $*p<0.05$; $**p<0.005$; $***p<0.001$; $****p<0.0001$. Data are presented as means ± SD unless otherwise indicated.

## Acknowledgements

We thank members of the White lab for valuable discussions and feedback on this project. We also thank the MSKCC flow cytometry core for assistance with cell sorting, Integrated Genomics Operation Core, funded by an NCI Cancer Center Support Grant (P30 CA08748) for sequencing assistance, and Molecular Cytology Core for their help with imaging. We thank the MSKCC aquatics core facility for their support of this work. RMW was funded through the NIH/NCI Cancer Center Support Grant P30 CA008748, the Melanoma Research Alliance, The Debra and Leon Black Family Foundation, NIH Research Program Grants R01CA229215 and R01CA238317, NIH Director's New Innovator Award DP2CA186572, Pershing Square Sohn Cancer Research Alliance, The Mark Foundation for Cancer Research, The American Cancer Society, The Alan and Sandra Gerry Metastasis Research Initiative at the Memorial Sloan Kettering Cancer Center, The Harry J Lloyd Foundation, Consano, and the Starr Cancer Consortium. This work was also supported by NIH ORIP R24OD020166 (MM). YM was supported by a Medical Scientist Training Program grant from the NIH under award number T32GM007739 to the Weill Cornell/Rockefeller/Sloan Kettering Tri-Institutional MD-PhD Program and Kirschstein-National Research Service Award (NRSA) predoctoral fellowship under award number F30CA265124. MVH was funded by a K99/R00 Pathway to Independence Award from the National Cancer Institute (1K99CA266931). SP was funded by a Molecular Imaging in Cancer Biology Training Grant at Memorial Sloan Kettering Cancer Center from the National Cancer Institute (T32CA254875) and the Ruth L Kirschstein National Research Service Award for Individual Predoctoral Fellows from the National Cancer Institute (F31CA271518).

## Additional information

### Competing interests

Maura McGrail: has competing interests with Recombinetics Inc, LifEngine Technologies, and LifEngine Animal Health Laboratories, Inc. Richard M White: is a paid consultant to N-of-One, a subsidiary of Qiagen, but this has no relationship with the work described in this manuscript. Senior editor, eLife. The other authors declare that no competing interests exist.

### Funding

| Funder | Grant reference number | Author |
| --- | --- | --- |
| National Cancer Institute | P30 CA008748 | Richard M White |

| Funder | Grant reference number | Author |
|---|---|---|
| Melanoma Research Alliance | | Richard M White |
| Debra and Leon Black Family Foundation | | Richard M White |
| National Institutes of Health | R01CA229215 | Richard M White |
| National Institutes of Health | R01CA238317 | Richard M White |
| National Institutes of Health | DP2CA186572 | Richard M White |
| Mark Foundation For Cancer Research | | Richard M White |
| American Cancer Society | | Richard M White |
| Alan and Sandra Gerry Metastasis and Tumor Ecosystems Center | | Richard M White |
| Harry J. Lloyd Charitable Trust | | Richard M White |
| National Institutes of Health | ORIP R24OD020166 | Maura McGrail |
| National Institutes of Health | T32GM007739 | Yilun Ma |
| National Cancer Institute | F30CA265124 | Yilun Ma |
| National Cancer Institute | 1K99CA266931 | Miranda V Hunter |
| National Cancer Institute | T32CA254875 | Sarah Perlee |
| National Cancer Institute | F31CA271518 | Sarah Perlee |
| Ludwig Institute for Cancer Research | Core Grant | Richard M White |
| Pershing Square Sohn Cancer Research Alliance | | Richard M White |
| Starr Foundation | | Richard M White |
| Consano | | Richard M White |

The funders had no role in study design, data collection and interpretation, or the decision to submit the work for publication.

## Author contributions

Sarah Perlee, Conceptualization, Formal analysis, Investigation, Visualization, Writing – original draft, Writing – review and editing; Yilun Ma, Zhitao Ming, Julia Xia, Investigation; Miranda V Hunter, Formal analysis, Investigation, Visualization, Writing – review and editing; Jacob B Swanson, Investigation, Visualization; Nelly M Cruz, Investigation, Writing – review and editing; Timothee Lionnet, Supervision; Maura McGrail, Supervision, Methodology; Richard M White, Conceptualization, Supervision, Funding acquisition, Writing – original draft, Writing – review and editing

## Author ORCIDs

Sarah Perlee https://orcid.org/0000-0003-3295-6755
Yilun Ma https://orcid.org/0000-0002-2297-9980
Miranda V Hunter https://orcid.org/0000-0002-6971-1738
Nelly M Cruz https://orcid.org/0000-0002-9040-6957
Timothee Lionnet https://orcid.org/0000-0003-1508-0202
Maura McGrail https://orcid.org/0000-0001-9308-6189
Richard M White https://orcid.org/0000-0001-9099-9169

## Ethics

All zebrafish experiments adhered to institutional animal protocols and were conducted in compliance with approved procedures. Fish stocks were maintained at a temperature of 28.5C, under 14:10 light:dark cycles, with pH set at 7.4, and salinity-controlled conditions. The zebrafish were fed a standard diet comprising brine shrimp followed by Zeigler pellets. Approval for the animal protocols outlined in this manuscript was obtained from the Memorial Sloan Kettering Cancer Center (MSKCC) Institutional Animal Care and Use Committee (IACUC), under protocol number 12-05-008.

Reviewer #1 (Public review): https://doi.org/10.7554/eLife.100257.3.sa1
Reviewer #3 (Public review): https://doi.org/10.7554/eLife.100257.3.sa2
Author response https://doi.org/10.7554/eLife.100257.3.sa3

# Additional files

## Supplementary files

MDAR checklist

Supplementary file 1. Sequences of primers, gRNAs, probes, and gBlock used in this study.

## Data availability

All data generated or analysed during this study are included in the manuscript and supplementary data.

The following previously published dataset was used:

| Author(s) | Year | Dataset title | Dataset URL | Database and Identifier |
|---|---|---|---|---|
| Wouters J, Kalender-Atak Z, Christiaens V, Spanier KI, Aerts S | 2020 | Single-cell analysis of gene expression variation and phenotype switching in melanoma | https://www.ncbi.nlm.nih.gov/geo/query/acc.cgi?acc=GSE134432 | NCBI Gene Expression Omnibus, GSE134432 |

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
