## [Editor Report · eLife Assessment]

This **important** manuscript introduces a genetic tool utilizing mutant mitfa-Cas9 expressing zebrafish to knockout genes to analyze melanocyte function in development and tumorigenesis. The data are **convincing** and the authors cover potential caveats from their model that might impact its utility for future work. This work significantly adds to the existing approaches in the field, as the mitfa:Cas9 strategy taken here provides a roadmap for generating similar platforms for using other tissue-specific regulators and Cas proteins in the future.

---

## [Referee Report · Reviewer #1 (Public review)]

Summary:

Perlee et al. sought to generate a zebrafish line where CRISPR-based gene editing is exclusively limited to the melanocyte lineage, allowing assessment of cell-type restricted gene knockouts. To achieve this, they knocked in Cas9 to the endogenous mitfa locus, as mitfa is a master regulator of melanocyte development. The authors use multiple candidate genes - albino, sox10, tuba1a, ptena/ptenb, tp53 - to demonstrate that their system induces lineage-restricted gene editing. This method allows researchers to bypass embryonic lethal and non-cell autonomous phenotypes emerging from whole body knockout (sox10, tuba1a), drive directed phenotypes, such as depigmentation (albino), and induce lineage-specific tumors, such as melanomas (ptena/ptenb, tp53, when accompanied with expression of BRAFV600E). The main weakness of the manuscript is that the mechanistic explanations proposed to underlie the presented phenotypes are minimally interrogated, but nonetheless interesting and motivating for future experimentation. Overall, there is a clear use for this genetic methodology, and its implementation will be of value to many in vivo researchers.

Strengths:

The strongest component of this manuscript is the genetic control offered by the mitfa:Cas9 system and the ability to make stable, lineage-specific knockouts in zebrafish. This is exemplified by the studies of tuba1a, where the authors nicely show non-cell autonomous mechanisms have obfuscated the role of this gene in melanocyte development. In addition, the mitfa:Cas9 system is elegantly straightforward and can be easily implemented in many labs. Mostly, the figures are clean, controls are appropriate, and phenotypes are reproducible. The invented method is a welcome addition to the arsenal of genetic tools used in zebrafish. The authors kindly and honestly responded to reviewer criticism, which has led to an improved manuscript and a pleasant review process.

Weaknesses:

The authors argue that the benefit of their system is the maintenance of endogenous regulatory elements. However, no direct comparison is made with other tools that offer similar genetic control, such as MAZERATI. This is a missed opportunity to provide researchers the ability to evaluate these two similar genetic approaches. There is a slight concern that tumor onset with this system is hindered by the heterozygous state it imparts to the lineage master regulator (here, mitfa). The authors do a good job at addressing these issues in the Discussion, but experimentation would have been appreciated. Additionally, the authors claim 86% of mitfa+ cells express Cas9. The image shown in Figure 1C does not do a convincing job at showing this percentage.

Another weakness of the manuscript regards minimally investigated mechanistic explanations for each biological vignette. Detailed mechanistic information is indeed out-of-scope for this manuscript, which intends to prove the efficacy of a genetic tool. Readers are cautioned to use the mechanistic insights from these vignettes as inspiration rather than bona fide truth.

The authors performed the necessary experiments to address each of the reviewers' concerns and thereby quell any substantial issues raised during the first review. They have additionally edited their language appropriately to make their claims more accurate. Their efforts during the review process are appreciated.

Conclusion:

The authors were highly receptive to reviewer comments and improved their manuscript from the first submission. The authors were successful in their goal of creating a rapid genetic approach to study cell-type specific genetic insults in vivo. They have presented multiple interesting and convincing stories to support the power of their invented methodology. The refined mechanisms underlying their observed phenotypes may be lacking but this does not take away from the methodological benefit this manuscript provides to the large field of in vivo researchers.

---

## [Referee Report · Reviewer #3 (Public review)]

Summary:

Perlee et al. present a method for generating cell-type restricted knockouts in zebrafish, focusing on melanocytes. For this method, the authors knock-in a Cas9 encoding sequence into the mitfa locus. This mitfaCas9 line has restricted Cas9 expression, allowing the authors to generate melanocyte-specific knockouts rapidly by follow-up injection of sgRNA expressing transposon vectors.

The paper presents some interesting vignettes to illustrate the utility of their approach. These include (1) a derivation of albino mutant fish as a demonstration of the method's efficiency, (2) an interrogation and novel description of tuba1a/tuba1c as a potential non-autonomous contributor to melanosome dispersion, and (3) the generation of sox10 deficient melanoma tumors that show "escape" of sox10 loss through upregulation of sox9. The latter two examples highlight the usefulness of cell-type targeted knockouts (Body-wide sox10 and tuba1a loss elicit developmental defects). Additionally, the tumor models involve highly multiplexed sgRNAs for tumor initiation which is nicely facilitated by the stable Cas9.

Strengths:

The approach is clever and could prove very useful for studying melanocytes and other cell types. As the authors hint at in their discussion, this approach would become even more powerful with the generation of other Cas9-restricted lineages so a single sgRNA construct can be screened across many lineages rapidly (or many sgRNA and fish lines screened combinatorially).

The biological findings used to demonstrate the power of the approach are interesting in their own right. The non-autonomous effect of tuba1a/tuba1c loss on melanosome dispersion are striking and demonstrates very nicely how one could use Perlee et al.'s approach to search for similar mechanisms systematically. The dual targeting nature of the tuba1a/tuba1c sgRNA also suggests similar approaches might be explored for knocking out paralogs. The observation of the sox9 escape mechanism with sox10 loss is a beautiful demonstration of the relevance of SOX10/SOX9's reciprocal regulation in vivo. This system would be a very nice model for further interrogating mechanisms/interventions surrounding Sox10 in melanoma.

Finally, the figure presentation is very nice. This work involves complex genetic approaches, including multiple fish generations and multiplexed construct injections. The vector diagrams and breeding schemes in the paper make everything very clear/"grok-able," and the paper was enjoyable to read.

Weaknesses:

The authors' claims are grounded and tested rigorously. The major weaknesses that we raised in the first round of reviews were either addressed experimentally or are now detailed as limitations in the text. Congrats on the beautiful paper!

---

## [Author Response]

The following is the authors’ response to the original reviews

**Public Reviews:**

**Reviewer #1 (Public review):**
Summary:Perlee et al. sought to generate a zebrafish line where CRISPR-based gene editing is exclusively limited to the melanocyte lineage, allowing assessment of cell-type restricted gene knockouts. To achieve this, they knocked in Cas9 to the endogenous mitfa locus, as mitfa is a master regulator of melanocyte development. The authors use multiple candidate genes - albino, sox10, tuba1a, ptena/ptenb, tp53 - to demonstrate their system induces lineagerestricted gene editing. This method allows researchers to bypass embryonic lethal and non-cell autonomous phenotypes emerging from whole body knockout (sox10, tuba1a), drive directed phenotypes, such as depigmentation (albino), and induce lineage-specific tumors, such as melanomas (ptena/ptenb, tp53, when accompanied with expression of BRAFV600E). While the genetic approaches are solid, the argued increase in efficiency of this model compared to current tools was untested, and therefore unable to be assessed. Furthermore, the mechanistic explanations proposed to underlie their phenotypes are mostly unfounded, as discussed further in the Weaknesses section. Despite these concerns, there is still a clear use for this genetic methodology and its implementation will be of value to many in vivo researchers.Strengths:The strongest component of this manuscript is the genetic control offered by the mitfa:Cas9 system and the ability to make stable, lineage-specific knockouts in zebrafish. This is exemplified by the studies of tuba1a, where the authors nicely show non-cell autonomous mechanisms have obfuscated the role of this gene in melanocyte development. In addition, the mitfa:Cas9 system is elegantly straightforward and can be easily implemented in many labs. Mostly, the figures are clean, controls are appropriate, and phenotypes are reproducible. The invented method is a welcomed addition to the arsenal of genetic tools used in zebrafish.Weaknesses:The major weaknesses of the manuscript include the overly bold descriptions of the value of the model and the superficial mechanistic explanations for each biological vignette.The authors argue that a major advantage of this system is its high efficiency. However, no direct comparison is made with other tools that achieve the same genetic control, such as MAZERATI. This is a missed opportunity to provide researchers the ability to evaluate these two similar genetic approaches. In addition, Fig.1 shows that not all melanocytes express Cas9. This is a major caveat that goes unaddressed. It is of paramount importance to understand the percentage of mitfa+ cells that express Cas9. The histology shown is unclear and too zoomed out of a scale to make any insightful conclusions, especially in Fig.S1. It would also be beneficial to see data regarding Cas9 expression in adult melanocytes, which are distinct from embryonic melanocytes in zebrafish. Moreover, this system still requires the injection of a plasmid encoding gRNAs of interest, which will yield mosaicism. A prime example of this discrepancy is in Fig.6, where sox10 is clearly still present in "sox10 KO" tumors.

We agree with these points. While our method has the advantage of endogenous knockin (thus keeping all regulatory elements), you are correct that we did not make a direct comparison with existing technologies like MAZERATI, and therefore we cannot make comparative claims about efficiency. Based on this, we have revised the manuscript to remove these points, reduce the strength/boldness of the claims, and make it more clear what our system achieves in comparison to existing systems. In reference to the other specific points you raise above about mosaicism and extent of Cas9 expression:

- We have added a paragraph to address the advantages and disadvantages of mitfaCas9 compared to expression of Cas9 with lineage-specific promoters including MAZERATI in the discussion.

- Figure 1C has been revised to more clearly show the overlap of mitfa and Cas9 in melanocytes.

- We then quantified the percentage of mitfa+ cells expressing Cas9 from the in situ hybridizations (Supplemental Figure S1D). We did attempt to look at Cas9 protein expression in both embryonic and adult melanocytes by immunofluorescence. Unfortunately, the Cas9 antibodies commercially available did not work on the zebrafish embryos or adult tailfins, so we are limited in proper quantification to the in situs in the embryos.

The authors argue that their model allows rapid manipulation of melanocyte gene expression. Enthusiasm for the speed of this model is diminished by minimal phenotypes in the F0, as exemplified in Fig.2. Although the authors say >90% of fish have loss of pigmentation, this is misleading as the phenotype is a very weak, partial loss. Only in the F1 generation do robust phenotypes emerge, which takes >6 months to generate. How this is more efficient than other tools that currently exist is unclear and should be discussed in more detail.

This needed clarification, and we have now modified the Discussion to reflect this more accurately. What we were trying to show is that both F0 and F1 fish can be useful in screening for the effect of any given gene. In the F0, while you are correct that the phenotype is indeed weak/partial, it is also quantifiable and therefore can be used as a rapid screen for potential effects of knockout, so it can help with speed. The major advantage of the F1 generation is that we can generate fully penetrant phenotypes for recessive genes since the fish just needs to have 1 copy of the Cas9/sgRNA instead of 2. This means we do not have to go to F2 or F3 generations, which really does save time. But we agree this could be achieved using MAZERATI, and so we have added these considerations to the manuscript, as we feel these are important.

In Figure 3, the authors find that melanocyte-specific knockout of sox10 leads to only a 25% reduction in melanocytes in the F1 generation. This is in contradiction to prior literature cited describing sox10 as indispensable for melanocyte development. In addition, the authors argue that sox10 is required for melanocyte regeneration. This claim is not accurate, as >50% of melanocytes killed upon neocuproine treatment can regenerate. This data would indicate that sox10 is required for only a subset of melanocytes to develop (Fig.3C) and for only a subset to regenerate (Fig.3G). This is an interesting finding that is not discussed or interrogated further.

We too were initially very puzzled by this result. We do not completely understand it, but we have two thoughts about it. First could be timing. sox10 usually starts to be expressed around the 1-somite stage, and so in the original sox10/colourless mutant (which truly has no melanocytes), sox10 will be lost during those early stages. In contrast, mitf comes on later (around 18hpf) so this might indicate that there is a subset of melanocytes that are dependent upon this early expression of sox10. This may indicate that there could be different functions of sox10 early in melanocyte development versus later timepoints after melanocytes have already been specified. This might also help explain our findings during regeneration. Second could be genetic compensation. Since in the other parts of the paper we seem to see a somewhat reciprocal relationship between sox10 and sox9, it is conceivable that loss of sox10 in the melanocytes could be compensated for by sox9 (or even other genes) in our CRISPR approach (as opposed to the ENU allele in colourless). Since we really do not fully understand this, we have added a section to the Discussion about this issue, mentioning these possibilities but leaving open other yet to be defined mechanisms.

Tumor induction by this model is weak, as indicated by the tumor curves in Figs.5,6. This might be because these fish are mitfa heterozygous. Whereas the avoidance of mitfa overexpression driven by other models including MAZERATI is a benefit of this system, the effect of mitfa heterozygosity on tumor incidence was untested. This is an essential question unaddressed in the manuscript.

We agree that in the BRAF;p53 group especially tumor incidence is very low, although PTEN loss does accelerate it. One possibility is exactly as you stated, and that mitfa heterozygosity is the etiology. The other possibility is that in the MAZERATI approach (https://pubmed.ncbi.nlm.nih.gov/30385465/) the authors used the casper background as opposed to the wild-type T5D as we did in our study. In unpublished observations, we have found that casper (with miniCoopR rescue) is markedly more sensitive to melanoma induction compared to WT fish in this setting. In fact, in looking at our BRAF;p53 curves compared to the original Patton paper curves (https://pubmed.ncbi.nlm.nih.gov/15694309/) which were also done in a WT background with no miniCoopR, they are fairly similar. This might indicate that casper + miniCoopR particularly sensitizes the fish to melanoma. However, because we do not fully know the reasons for this, we have now included both of these possible reasons in the Discussion.

In Fig.6, the authors recapitulate previous findings with their model, showing sox10 KO inhibits tumor onset. The tumors that do develop are argued to be highly invasive, have mesenchymal morphology, and undergo phenotypic switching from sox10 to sox9 expression. The data presented do not sufficiently support these claims. The histology is not readily suggestive of invasive, mesenchymal melanomas. Sox10 is still present in many cells and sox9 expression is only found in a small subset (<20%). Whether sox10-null cells are the ones expressing sox9 is untested. If sox9-mediated phenotypic switching is the major driver of these tumors, the authors would need to knockout sox9 and sox10 simultaneously and test whether these "rare" types of tumors still emerge. Additional histological and genetic evaluation is required to make the conclusions presented in Fig.6. It feels like a missed opportunity that the authors did not attempt to study genes of unknown contribution to melanoma with their system.

We did not mean to overstate the admittedly early observations from these fish. Invasiveness in the fish models can be difficult to precisely quantify, and therefore is somewhat qualitative. While we did not mean to imply that every cell that loses sox10 will become sox9 positive (which is clearly not the case), the human single-cell RNA-seq data does suggest these are somewhat mutually exclusive populations (https://pubmed.ncbi.nlm.nih.gov/32753671/). This phenomenon has also long been observed even prior to single-cell approaches (https://pubmed.ncbi.nlm.nih.gov/25629959/). So while we agree our data is not definitive in this regard, it is consistent with the literature and was presented mainly to provide areas for future exploration with the model.

Overall, this manuscript introduces a solid method to the arsenal of zebrafish genetic tools but falls short of justifying itself as a more efficient and robust approach than what currently exists. The mechanisms provided to explain observed phenotypes are tenuous. Nonetheless, the mitfa:Cas9 approach will certainly be of value to many in vivo biologists and lays the foundation to generate similar methods using other tissue-specific regulators and other Cas proteins.

We hope that by toning down the language around what we have observed, and providing as honest an assessment as possible as to what might be occurring, that the manuscript will be helpful for future studies aiming to knock out genes in the melanocyte lineage.

**Reviewer #2 (Public review):**
Summary:This manuscript describes a genetic tool utilizing mutant mitfa-Cas9 expressing zebrafish to knockout genes to analyze their function in melanocytes in a range of assays from developmental biology to tumorigenesis. Overall, the data are convincing and the authors cover potential caveats from their model that might impact its utility for future work.Strengths:The authors do an excellent job of characterizing several gene deletions that show the specificity and applicability of the genetic mitfa-Cas9 zebrafish to studying melanocytes.Weaknesses:Variability across animals not fully analyzed.

To more clearly show variability across animals, we calculated the percentage of mitfa+ cells that express Cas9 across n=7 mitfaCas9 embryos. We also expanded Supplemental Figure 2 to show loss of pigmentation across n=7 individual adult MG-*albino* F2 fish instead of one representative image.

**Reviewer #3 (Public review):**
Summary:Perlee et al. present a method for generating cell-type restricted knockouts in zebrafish, focusing on melanocytes. For this method, the authors knock-in a Cas9 encoding sequence into the mitfa locus. This mitfaCas9 line has restricted Cas9 expression, allowing the authors to generate melanocyte-specific knockouts rapidly by follow-up injection of sgRNA expressing transposon vectors.The paper presents some interesting vignettes to illustrate the utility of their approach. These include (1) a derivation of albino mutant fish as a demonstration of the method's efficiency, (2) an interrogation and novel description of tuba1a as a potential non-autonomous contributor to melanocyte dispersion, and (3) the generation of sox10 deficient melanoma tumors that show "escape" of sox10 loss through upregulation of sox9. The latter two examples highlight the usefulness of cell-type targeted knockouts (Body-wide sox10 and tuba1a loss elicit developmental defects). Additionally, the tumor models involve highly multiplexed sgRNAs for tumor initiation which is nicely facilitated by the stable Cas9.

Strengths:

The approach is clever and could prove very useful for studying melanocytes and other cell types. As the authors hint at in their discussion, this approach would become even more powerful with the generation of other Cas9-restricted lineages so a single sgRNA construct can be screened across many lineages rapidly (or many sgRNA and fish lines screened combinatorially).The biological findings used to demonstrate the power of the approach are interesting in their own right. If it proves true, tuba1a's non-autonomous effects on melanosome dispersion are striking, and this example demonstrates very nicely how one could use Perlee et al.'s approach to search for other non-autonomous mechanisms systematically. Similarly, the observation of the sox9 escape mechanism with sox10 loss is a beautiful demonstration of the relevance of SOX10/SOX9's reciprocal regulation in vivo. This system would be a very nice model for further interrogating mechanisms/interventions surrounding Sox10 in melanoma.Finally, the figure presentation is very nice. This work involves complex genetic approaches including multiple fish generations and multiplexed construct injections. The vector diagrams and breeding schemes in the paper make everything very clear/"grok-able," and the paper was enjoyable to read.Weaknesses:The mitfa-driven GFP on their sgRNA-expressing cassette is elegant, but it makes one wonder why the endogenous knock-in is necessary. It would strengthen the motivation of the work if the authors could detail the potential advantages and disadvantages of their system compared to expressing Cas9 with a lineage-specific promoter from a transposon in their introduction or discussion.

We agree this needed a better and more clear explanation. There are many excellent examples of promoter driven Cas9 approaches. Within melanocytes, Ablain and others have developed the MAZERATI system (https://pubmed.ncbi.nlm.nih.gov/30385465/) which is very powerful, especially for melanoma development. In our minds, the major advantage of endogenous knockin is that we retain all of the natural regulatory elements (many of which are not known) and so small promoter fragments always run the risk of missing certain types of regulation. While these regulatory elements may not matter under homeostatic conditions, they may become very important under perturbation, stress or disease states. This is why it is common, for example, in the mouse field, to knock in things like Cre into endogenous loci. We have now added a clarification of this to the manuscript.

Related to the above - is mitfa haplosufficient? If the mitfaCas9/+ fish have any notable phenotypes, it would be worth noting for others interested in using this approach to study melanoma and pigmentation.

In normal melanocytes, *mitfa* is haplosufficient. There are no visible differences between mitfaCas9/+ and wild-type fish at any stages of development (Figure S1C). Although we did not directly compare tumor growth in mitfa-/+ and *mitfa*+/+ fish in this study, it is possible that the disruption of *mitfa* in mitfaCas9/+ fish affects melanoma development. Most zebrafish melanoma models involve the overexpression of *mitfa* with MiniCoopR vectors and it would be interesting in future studies to determine how *mitfa* heterozygosity affects melanoma initiation or progression.

A core weakness (and also potential strength) of the system is that introduced edits will always be non-clonal (Fig 2H/I). The activity of individual sgRNAs should always be validated in the absence of any noticeable phenotype to interpret a negative result. Additionally, caution should be taken when interpreting results from rare events involving positive outgrowth (like tumorogenesis) to account for the fact many cells in the population might not have biallelic null alleles (i.e., 100% of the gene product removed).Along those lines: in my opinion, the tuba1a results are the most provocative finding in the paper, but they lack key validation. With respect to cutting activity, the Alt-R and transgenic sgRNA expression approaches are not directly comparable. Since there is no phenotype in the melanocyte specific tuba1a knockouts, the authors must confirm high knockout efficiency with this set of reagents before making the claim there is a non-autonomous phenotype. This can be achieved with GFP+ sorting and NGS like they performed with their albino melanocytes.The whole-body tuba1a knockout phenotype is expected to be pleiotropic, and this expectation might mask off-target effects. Controls for knockout specificity should be included. For instance, confidence in the claims would greatly increase if the dispersed melanosome phenotype could be recovered with guide-resistant tuba1a re-expression and if melanocyte-restricted tuba1a reexpression failed to rescue. As a less definitive but adequate alternative, the authors could also test if another guide or a morpholino against tuba1a phenocopies the described Alt-R edited fish.

Thank you for your thoughtful suggestions, which led us to an important discovery. While validating the original *tuba1a* guide RNA, we found that *tuba1a* sg1 also targets *tuba1c*, a gene that shares 99.78% homology with *tuba1a* in zebrafish. To determine which gene was responsible for the melanocyte phenotype, we designed multiple new guide RNAs specifically targeting either *tuba1a* or *tuba1c* and used Alt-R to globally knock them out in zebrafish embryos. However, none of these guides successfully replicated the phenotype (Sanger sequencing validation for the most efficient *tuba1a* and *tuba1c* guides is provided below).

Ultimately, we identified a new guide RNA (5’-GGTCTACAAAGACAGCCCTA-3’) that successfully phenocopied the original *tuba1a* sg1 melanocyte phenotype. *Tuba1c*—but not *tuba1a*—was predicted to have a mismatch at the 3’ end of the guide sequence, which is typically expected to inhibit target cleavage. Surprisingly, despite this mismatch, we observed robust cleavage in both *tuba1a* and *tuba1c*. Since the melanocyte phenotype was only reproducible when *both tuba1a* and *tuba1c* were targeted, this suggests potential compensatory interactions between these highly similar genes. We have updated the text and figures to reflect this finding and have included validation of this second guide RNA (*tuba1a/c* sg2) in Supplemental Figure 3.

As you suggested, we also conducted GFP+ sorting and NGS to confirm knockout of both *tuba1a* and *tuba1c* in melanocytes of mitfaCas9 fish (Figure S3G). The knockout percentages were comparable to those observed in our previous experiment with MG_-albino_ fish. This also confirms that this method can be used to sort and sequence GFP+ cells even when pigmentation is retained, which was not the case for albino fish.

I have similar questions about the sox10 escapers, but these suggestions are less critical for supporting the authors claims (especially given the nice staining). Are the sox10 tumors relatively clonal with respect to sox10 mutations? And are the sox10 tumor mutations mostly biallelic frameshifts or potential missense mutations/single mutations that might not completely remove activity? I am particularly curious as SOX10 doesn't seem to be completely absent (and is still very high in some nuclei) in the immunohistochemistry.

We attempted to address this question by performing DNA sequencing on the FFPE blocks that we had retained from the original study. While our sequencing facility said this should be possible, we could not consistently generate high enough quality DNA to make a definitive statement either way. While we are very curious to know what the nature of the mutations are in these “escapers”, the student who performed these studies has now graduated, and it would take us several additional months to a year to fully address it. Given this, we would prefer to leave this open question to a future paper, but have addressed this limitation in the Discussion.

**Recommendations for the authors:**

**Reviewing Editor:**
Overall, the reviewers felt and eLife concurs that your manuscript is insightful and appropriate for publication. Reviewers were impressed by your generating a zebrafish line where CRISPRbased gene editing is exclusively limited to the melanocyte lineage, allowing assessment of celltype restricted gene knockouts. Your use of multiple candidate genes to demonstrate that your system induces lineage-restricted gene editing is compelling and will be of interest to the broad readership of eLife. This method will allow researchers to bypass embryonic lethal and non-cell autonomous phenotypes emerging from whole body knockout, drive directed phenotypes, such as depigmentation, and induce lineage-specific tumors, such as melanomas. This said, the argued increase in efficiency of this model compared to current tools was untested, and therefore it remains difficult for a reader to assess the extent to which your new model represents a major advance over prior ones. Of additional concern are the mechanistic explanations proposed to underlie the phenotypes, as these are largely unfounded. Thus, in preparing your final publication version of the paper, eLife strongly encourages you to fully address the reviewers' thoughtful comments. In particular, the boldness of the claims made in the manuscript should be reduced. Terms like "highly efficient" and "rapid" are unsupported due to the lack of comparison with other well-established methods, like MAZERATI.

As discussed above in each of the reviewer points above, we agree with both of these points. We have reduced the boldness of the claims, with a better discussion of the different approaches. We also address the potential mechanisms of our observations, and where and why we still lack an understanding of what gives rise to those phenotypes.

There are also some minor discrepancies that should be edited in the manuscript: Fig.2A plasmid description is written oppositely in text; Fig.3 labels G-H are swapped in the legend description; Fig.5A MTdT is unexplained. This is a non-exhaustive list, and the authors are encouraged to carefully read through their manuscript to revise other minor mistakes and formatting errors.

Figure 2A was revised to show the correct orientation of mitfa:GFP and the guide RNA cassette as described in the text. Figure 3 legend was fixed. We have gone through the manuscript again to make sure we have not made any other errors, to the best of our knowledge.

The biggest concern is the expression of cas9 and the weak histological support shown in Fig.1 and Fig.S1. It would be a benefit to all readers and potential future users to know how robust cas9 expression is in the melanocyte lineage. It would be helpful if there is a way to analyze the percentage of cells that are mutated in each animal to understand the variability that can exist across animals with the method.

We have revised Figure 1C to show additional melanocytes and added a new quantification of Cas9 RNA expression in melanocytes (S1D).

The analysis of the scRNA sequencing could also be described more fully.

More details have been added to the scRNA sequencing analysis including the functions that were used.

The final major concern is whether this model is genuinely more valuable than MAZERATI. A more elaborate discussion would benefit potential future users to guide their decisions regarding which tool best suits their experimental goals.

As noted above, we agree with this statement. The reviewers are correct in that we did not directly compare our system to MAZERATI, and therefore cannot make any claims about efficiency in a comparative regard. Therefore, in our revised Discussion, we talk about the relative strengths and weaknesses of each approach, and emphasize that our approach mainly has the advantage of retaining endogenous regulatory elements for *mitfa*, but that each user should decide which is the best approach for their problem.

There are also some minor concerns that should be addressed.Are the mitfaCas9 fish used as homozygotes before the first cross? If so, might be nice to include their nacre-like phenotype in diagrams like Fig 2A.

For these studies, heterozygous mitfaCas9 fish were used for all breedings and progeny were sorted for BFP+ eyes. This enabled the comparison to sibling controls without Cas9 expression.

BFP+ eye screening for mitfaCas9 is elegant and included nicely in the diagrams. Are germline sgRNA integrants identified in F1 with melanocyte GFP? Or present at a high enough efficiency that this is not relevant? This would be good to include in the diagrams.

Germline sgRNA integrants are identified with melanocyte GFP in embryos. Figure 2A has been edited to show GFP expression.

Most cells are GFP positive in S3C (the F0 "mosaic"). It might be nice to show a single GFP stripe like in the other panels for direct comparison of edited/non-edited in the same fish.

This figure (now S3E) has been edited to show a clear comparison between GFP+ and GFP- cells in the same fish.

177 - CRISPR-Seq is basically amplicon sequencing. This would measure efficiency but not "specificity" as described. Off-target activity would have to be measured at other loci etc. Not necessary to do, but I don't think measured.

In this case, “specificity” refers to cell type specificity, not genomic specificity. We are measuring cell type specificity by comparing on-target cutting in GFP+ cells (melanocytes) versus GFP- cells (non-*mitfa* expressing cells). We did not look at off-target activity of Cas9 in this study and have edited the text to make this clearer.

219 -"several gaps were visible"

Fixed

286 - TUBA1A should be italicized

Fixed

399 - SOX9's most enriched dependency in DepMap is cutaneous melanoma and its top coessential gene is SOX10. I'm not sure the SOX9/SOX10 interaction couldn't be parsed from DepMap alone.

This is true, and the DepMap was actually somewhat of an inspiration for our own studies. We have modified the line to acknowledge this and explain the main advantage of our system is in vivo confirmation of what the DepMap had alluded to.

433 - "fewer animals since all F1 animals (even those for recessive alleles) are informative."The fact that this is approach is faster and more efficient per animal is important to highlight (and very believable), but is this technically true given not all F1 fish will have Cas9 or a germline sgRNA integration?

In considering this statement, we agree with you and decided to remove it from the text.

We hope the comments in both the public and private reviews will help improve the manuscript.
**Reviewer #1 (Recommendations for the authors):**
Overall, the boldness of the claims made in the manuscript should be reduced. Terms like "highly efficient" and "rapid" are unsupported due to the lack of comparison with other wellestablished methods, like MAZERATI.

As discussed above, we agree with this and have now modified the manuscript to better reflect what our system achieves in comparison to the well developed systems such as MAZERATI. Because we have not done a direct comparison, we are not able to make any claims about comparative efficiency, and instead focus on the potential benefits of a knockin approach, which is the maintenance of endogenous regulatory elements.

There are some minor discrepancies that should be edited in the manuscript: Fig.2A plasmid description is written oppositely in text; Fig.3 labels G-H are swapped in the legend description; Fig.5A MTdT is unexplained. This is a non-exhaustive list, and the authors are encouraged to carefully read through their manuscript to revise other minor mistakes and formatting errors.

Figure 2A was revised to show the correct orientation of mitfa:GFP and the guide RNA cassette as described in the text. Figure 3 legend was fixed. We have gone through the manuscript again to make sure we have not made any other errors, to the best of our knowledge.

The biggest concern is the expression of cas9 and the weak histological support shown in Fig.1 and Fig.S1. It would be a benefit to all readers and potential future users to know how robust cas9 expression is in the melanocyte lineage.

We have revised Figure 1C to show additional melanocytes and added a new quantification of Cas9 RNA expression in melanocytes (S1D).

The second major concern is whether this model is genuinely more valuable than MAZERATI. A more elaborate discussion would benefit potential future users to guide their decision regarding which tool best suits their experimental goals.

As noted above, we agree with this statement. The reviewers are correct in that we did not directly compare our system to MAZERATI, and therefore cannot make any claims about efficiency in a comparative regard. Therefore, in our revised Discussion, we talk about the relative strengths and weaknesses of each approach, and emphasize that our approach mainly has the advantage of retaining endogenous regulatory elements for mitfa, but that each user should decide which is the best approach for their problem.

We hope the comments in both the public and private reviews will help improve the manuscript.
**Reviewer #2 (Recommendations for the authors):**
While that authors show the indel charts for the Crispr mutations generated in the supplement. However, I wonder if there is a way to analyze the percentage of cells that are mutated in each animal to understand the variability that can exist across animals with the method.

We have revised Figure 1C to show additional melanocytes and added a new quantification of Cas9 RNA expression in melanocytes (S1D).

The analysis of the scRNA sequencing could be described more fully.

More details have been added to the scRNA sequencing analysis including the functions that were used.

**Reviewer #3 (Recommendations for the authors):**
This was an excellent read, and I'm very interested in seeing it in its final form. Congratulations! My larger critiques are outlined in the public reviews. A few smaller points:Are the mitfaCas9 fish used as homozygotes before the first cross? If so, might be nice to include their nacre-like phenotype in diagrams like Fig 2A.

For these studies, heterozygous mitfaCas9 fish were used for all breedings and progeny were sorted for BFP+ eyes. This enabled the comparison to sibling controls without Cas9 expression.

BFP+ eye screening for mitfaCas9 is elegant and included nicely in the diagrams. Are germline sgRNA integrants identified in F1 with melanocyte GFP? Or present at a high enough efficiency that this is not relevant? This would be good to include in the diagrams.

Germline sgRNA integrants are identified with melanocyte GFP in embryos. Figure 2A has been edited to show GFP expression.

Most cells are GFP positive in S3C (the F0 "mosaic"). It might be nice to show a single GFP stripe like in the other panels for direct comparison of edited/non-edited in the same fish.

This figure (now S3E) has been edited to show a clear comparison between GFP+ and GFP- cells in the same fish.

177 - My understanding is that CRISPR-Seq is basically amplicon sequencing. This would measure efficiency but not "specificity" as described. Off-target activity would have to be measured at other loci etc. Not necessary to do in my opinion, but I don't think measured.

In this case, “specificity” refers to cell type specificity, not genomic specificity. We are measuring cell type specificity by comparing on-target cutting in GFP+ cells (melanocytes) versus GFP- cells (non-*mitfa* expressing cells). We did not look at off-target activity of Cas9 in this study and have edited the text to make this clearer.

219 -"several gaps were visible"

Fixed

286 - TUBA1A should be italicized

Fixed

399 - I think I understand the logic of the DepMap argument, and the importance of studying tumor initiation in vivo stands for itself. But here is maybe not the best example (or might need clarification)? - SOX9's most enriched dependency in DepMap is cutaneous melanoma and its top co-essential gene is SOX10. I'm not sure the SOX9/SOX10 interaction couldn't be parsed from DepMap alone.

This is true, and the DepMap was actually somewhat of an inspiration for our own studies. We have modified the line to acknowledge this and explain the main advantage of our system is in vivo confirmation of what the DepMap had alluded to.

433 - "fewer animals since all F1 animals (even those for recessive alleles) are informative."The fact that this is approach is faster and more efficient per animal is important to highlight (and very believable), but is this technically true given not all F1 fish will have Cas9 or a germline sgRNA integration?

In considering this statement, we agree with you and decided to remove it from the text.